# Carbon isotopes of dissolved inorganic carbon reflect utilization of different carbon sources by microbial communities in two limestone aquifer assemblages

Martin E. Nowak[1], Valérie F. Schwab[2], Cassandre S. Lazar[3], Thomas Behrendt[1], Bernd Kohlhepp[2],  Kai Uwe Totsche[2], Kirsten Küsel[3,4], Susan E. Trumbore[1]

[1]Department for Biogeochemical Processes, Max-Planck Institute for Biogeochemistry, Hans-Knöll Straße 10, 07745 Jena, Germany
[2]Chair of Hydrogeology, Institute of Geosciences, Friedrich Schiller University Jena, Germany
[3]Aquatic Geomicrobiology, Institute of Ecology, Friedrich Schiller University Jena, Dornburger Str. 159, 07743 Jena, Germany
[4] German Centre for Integrative Biodiversity Research (iDiv), Halle-Jena-Leipzig, Leipzig, Germany

*Correspondence to*: Martin E. Nowak (mnowak@bgc-jena.mpg.de)

**Abstract.**  Isotopes of dissolved inorganic carbon (DIC) are used to indicate both transit times and biogeochemical evolution of groundwaters. These signals can be complicated in carbonate aquifers, as both abiotic (i.e. carbonate equilibria) and biotic factors influence $\delta^{13}C$ and $^{14}C$ of DIC. We applied a novel graphical method for tracking changes in $\delta^{13}C$ and $^{14}C$ of DIC in two distinct aquifer complexes identified in the Hainich Critical Zone Exploratory (CZE), a platform to study how water

transport links surface and shallow groundwaters in limestone and marlstone rocks in central Germany. For more quantitative estimates of contributions of different biotic and abiotic carbon sources to the DIC pool, we used the geochemical modelling program NETPATH, which accounts for changes in dissolved ions in addition to C isotopes.

Although water residence times in the Hainich CZE aquifers based on hydrogeology are relatively short (years or less), DIC isotopes in the shallow, mostly anoxic, aquifer assemblage (HTU) were depleted in $^{14}C$ compared to a deeper, oxic, aquifer

complex (HTL). Carbon isotopes and chemical changes in the deeper HTL wells could be explained by interaction of recharge waters equilibrated with post-bomb $^{14}C$ sources with carbonates. However, oxygen depletion and $\delta^{13}C$ and $^{14}C$ values of DIC below those expected from the processes of carbonate equilibrium alone indicate considerably different biogeochemical evolution of waters in the upper aquifer assemblage (HTU wells). Changes of $^{14}C$ and $^{13}C$ in the upper aquifer complexes result from a number of biotic and abiotic processes, including oxidation of $^{14}C$ depleted OM derived

from recycled microbial carbon and sedimentary organic matter as well as water rock interactions. The microbial pathways inferred from DIC isotope shifts and changes in water chemistry in the HTU wells were supported by comparison with *in situ* microbial community structure based on 16S rRNA analyses.

Our findings demonstrate the large variation in the importance of biotic as well as abiotic controls on $^{13}$C and $^{14}$C of DIC in closely related aquifer assemblages. Further, they support the importance of subsurface derived carbon sources like DIC for chemolithoautotrophic microorganisms as well as rock-derived organic matter for supporting heterotrophic groundwater microbial communities and indicate that even shallow aquifers have microbial communities that use a variety of subsurface derived carbon sources.

## 1    1 Introduction

Groundwater is the most important freshwater reserve on earth and a crucial part of the global hydrological cycle. Although the proportion of groundwater on global freshwater reserves is only 0.06 %, it represents as much as 98 % of readily available water for humans, livestock, and agriculture.

According to the Intergovernmental Panel on Climate Change (IPCC), groundwater demand by humans is likely to increase in future, due to a general increase of global water use and a decline in surface water availability caused by higher precipitation variability (Parry et al., 2007). In contrast to expected higher groundwater withdrawal, groundwater recharge rates are likely to decrease on a regional scale because of climate change (Aeschbach-Hertig and Gleeson, 2012).

The Critical Zone (CZ) is defined as the space ranging from the outer extent of vegetation through soils, down to the saturated and unsaturated bedrock (NRC, 2001). It is the crucial connection between groundwater and surface conditions and the space where fundamental physical, chemical and biological processes act that are of high importance for sustaining soil and groundwater quality for agricultural and groundwater use (Akob and Küsel, 2011). Assessments of groundwater vulnerability and sustainable groundwater management require a sound knowledge of water movement and carbon transport through the CZ (Küsel et al., 2016).

In this study, we measured radiocarbon and stable carbon isotopes of dissolved inorganic carbon (DIC), dissolved organic carbon (DOC) and particulate organic carbon (POC) in a transect of wells located in the Hainich Critical Zone Exploratory (Hainich CZE), Thuringa, Central Germany. Our aim was to use the isotopic composition of all carbon species in the groundwater as a proxy to study both, carbon turnover and water dynamics in two superimposed limestone aquifer assemblages with different flow dynamics and physicochemical properties (Fig. 1).

Radiocarbon ($^{14}$C) and stable carbon isotopes ($^{13}$C/$^{12}$C) of dissolved constituents provide a useful tool to trace water and carbon (C) movement through the critical zone, as well as to identify different sources contributing to the aquifers' carbon pools (Bethke and Johnson, 2008). The most common approach in applying radiocarbon in groundwater studies is measuring the $^{14}$C activity of dissolved inorganic carbon (DIC), which includes dissolved $CO_2$, bicarbonate and carbonate ions. These C

species are derived from equilibration of percolating waters with soil atmosphere $CO_2$, as well as equilibration of the dissolved $CO_2$ with carbonates in the soil matrix or aquifer rocks.

In order to determine initial [14]C concentrations in DIC, numerous correction models have been applied to account for the different processes that affect DIC and bias [14]C ages (Han and Plummer, 2016). Corrections have to be considered for dissolution of carbonates by dissolved $CO_2$ (Tamers et al., 1975) according to:

$$CO_2 + H_2O + CaCO_3 \rightarrow Ca^{2+} + 2HCO_3^- \tag{1}$$

, isotopic exchange between DIC and soil $CO_2$ (Fontes, 1992;Han and Plummer, 2013):

$$C^*O_{2(g)} + HCO_3^- \leftrightarrow CO_{2(g)} + HC^*O_3^- \tag{2}$$

, as well as isotopic exchange between DIC and carbonates in the aquifer (Eichinger, 1983;Fontes and Garnier, 1979;Han and Plummer, 2013):

$$HC^*O_3^- + CaCO_{3(s)} \leftrightarrow HCO_3^- + CaC^*O_{3(s)} \tag{3}$$

where * in equations two and three refers to carbon atoms that were exchanged between gaseous liquid and solid phases. Other factors influencing [13]C and [14]C of DIC are heterotrophic respiration of organic matter (OM) (Aravena et al., 1995) and mineral precipitation and weathering (Wigley, 1976).

Han et al. (2012) developed a novel graphical method, the Han-Plummer plot, that allows easy recognition of systematic relationships between [13]$C_{DIC}$ and [14]$C_{DIC}$, and are indicative for the processes described above. For example, interactions between carbonates and DIC should follow a mixing line, as [13]C and [14]C are affected in constant proportions set by the carbonate endmember. On the other hand addition of organic matter C can vary in both the [13]C signature (e.g. fermentation versus oxidation by $O_2$) and radiocarbon signature of added C, depending on the organic matter source.

In limestone-landscapes like the Hainich CZE, groundwater recharge rates can be fast due to karstification or slower due to thick soils developed on quarternary loess deposits (Kohlhepp et al., 2016). In such regions, recharge waters containing biogenic soil $CO_2$ that is mostly of recent origin near the surface react with the carbonates in aquifer rock according to the

stoichiometry of Eq. (1). The DIC in such waters fall within a very specific region on the Han-Plummer plot, the so-called Tamer´s point (Han et al., 2012), i.e. with 50% of the DIC derived from the soil-$CO_2$ and 50% from the carbonate rock according to Eq. (1). Values that fall off this pure calcite equilibration point reflect the influence of isotopic exchange in the aquifer or soil according to Eq. (2) and (3), i.e. water-carbonate rock interactions, or microbial oxidation of organic matter,

respectively. The relative importance of these processes can be distinguished according to the specific position of DIC carbon isotopic values in the Han-Plummer plot (Han et al., 2012).

Carbon isotopes of DIC can give therefore information about both, movement of water through the soil (i.e. information about the signature of the $CO_2$ in the unsaturated zone; Gillon et al., 2012), as well as sources and sinks of different carbon pools within the aquifer (Aravena et al., 1995).

While the graphical method emphasizes only C isotopes, information from other dissolved constituents can provide additional information about the biogeochemical factors influencing groundwater. For example, carbonate dissolution will not only affect DIC but also the concentrations of dissolved $Ca^{+2}$ and $Mg^{+2}$ and microbial processes like sulphate reduction will alter $SO_4^{2-}$ concentrations. In order to assess quantitatively the contribution of biotic and abiotic processes to groundwater carbon biogeochemistry and their impact on DIC isotopes, we used the geochemical inverse modelling program

NETPATH (Plummer et al., 1994) that takes alterations in water chemistry into account.

The aim of our study was to use carbon isotopes of all accessible organic and inorganic carbon species in our studied limestone aquifers and use them to elucidate water flow as well as carbon-turnover by applying latest graphical and computational methods including water chemistry and microbiology. The carbon turnover part, especially transformation and formation of organic matter by microorganisms and their interaction with the aquifer rock and water chemistry was a

main target of our study. A special focus of our study was to evaluate the contribution of autotrophic microorganisms to carbon cycling within the aquifer. Chemolithoautotrophic microorganisms, i.e. microbes that metabolise $CO_2$ instead of organic carbon, have been shown to be key players and important primary producers in groundwater microbial communities (Alfreider et al., 2012; Hutchins et al., 2016; Kellermann et al., 2012). A high potential for microbial $CO_2$ fixation has already been demonstrated in our studied aquifers by molecular analyses (Herrmann et al., 2015; Lazar et al., 2016a; Schwab

et al., submitted). We hypothesized that turnover of OM derived from chemoautotrophic microorganisms should be reflected in DIC isotopes, since OM derived from $CO_2$ fixation should be isotopically distinct from other sources like surface derived OM or sedimentary organic matter. Therefore, we conducted 16S rRNA gene assays, in order to determine the microbial community structure within the two limestone aquifer assemblages and relate it to measured carbon isotopes of DIC, DOC and POC as well as water chemistry.

## 2    Methods

### 2.1    Study site

The Hainch CZE is located in Thuringia, Central Germany. It spreads from the Hainich low mountain range (in the SW), representing the groundwater recharge area of this study, towards the valley of the Unstrut river (in the NE). The southwestern part of the study site shares the largest deciduous beech forest in Germany, the Hainich National Park. Within the forest area, the NW-SE oriented Hainich ridge is the topographical and subsurface water divide with surface/subsurface discharge towards the east (Unstrut subcatchment) and west (Werra subcatchment). The geological succession of Mesozoic sedimentary rocks is moderately inclined towards the NE and comprises the Muschelkalk (m) group outcropping in the upper and midslope area and the Keuper (k) group at the footslope. As the strata dips steeper than the slope angle, lower stratigraphic units outcrop in higher topographic positions. The Upper Muschelkalk (mo) subgroup, which hosts the aquifer assemblages of this study, is further subdivided into the Trochitenkalk formation (moTK) with predominantly limestones and the alternated bedded limestone-marlstone succession of the Meissner formation (moM). Mesozoic rocks are partly to totally covered by Pleistocene Loess loam in the mid/footslope area. Footslope valleys are filled with unconsolidated alluvium (Küsel et al., 2016; Kohlhepp et al., 2016). Agricultural areas with different management intensities surround the largely unmanaged forest area at the eastern hillslope of the Hainich low mountain range.

The Hainich CZE comprises here a number of surface and belowground observational plots along a 5.4 km-long hillslope transect in an intensively investigated area of about 29 km$^2$ (Kohlhepp et al., 2016; Küsel et al., 2016). A groundwater well transect consisting of five locations (H1 (upslope) to H5 (toeslope)) that also span a land-use gradient from deciduous managed forest (H1), unmanaged woodland (H2), grassland/pasture (H3) to cropland agriculture (H4 and H5) longitudinal to the assumed ground water flow direction (Fig. 1). The wells provide access to two main aquifer assemblages: referred to as HTL and HTU (Hainich transect lower/upper aquifer assemblage, respectively (Küsel et al., 2016), which consist of one (HTL) and, respectively, nine (HTU) individual aquifer storeys (Küsel et al., 2016;Kohlhepp et al., 2016). The HTU aquifers are sampled at depths ranging between 0.6 m (mid-slope; H3) and 54 m (downslope; H5), while the HTL aquifers are sampled at 2 m (upslope;H1) and 89 m (H5) (Fig. 1). HTL comprises a complex of aquifer layers in the Trochitenkalk formation (moTK) and comprises thickly bedded porous limestone packages that act as karst-fracture aquifers. HTU comprises mainly an layered aquifer complex within the overlying Meissner formation (moM), comprising fracture aquifers with fine fissures and less pronounced karstification due to the finely alternating succession of limestones and low-permeable marlstone beds (Fig. 1; Küsel et al., 2016; Kohlhepp et al., 2016). Vertical exchange is strongly inhibited by frequent and low-permeable marlstone interbeds resulting in confined flow conditions and a layer-cake architecture (Kohlhepp et al., 2016). HTL is characterised by more intense karstification and fast flow through large conduits (Küsel et al., 2016). Both aquifer assemblages and the included aquifer storeys are separated by intercalated marlstone interbeds.

Groundwater recharge in HTL takes place mainly within the forested upper to middle hillslope of the transect. By contrast, HTU outcrops cover all land use types and recharge also takes place in the lower hill slope area, with mixed land use with forests, pastures and cropland areas (Küsel et al., 2016) (Fig. 1).

Soils in the forested upper to midslope area are predominantly shallow Rendzic Leptosols and Cambisols, that developed mainly on marl- and limestones. Luvisols and Planosols/Stagnosols (WRB nomenclature; WRB, 2006) developed on siliciclastic sediments like Pleistocene Loess loam or unconsolidated Holocene deposits at the foot slope position (Küsel et al., 2016).

Groundwater samples analysed in this study were obtained from 8 groundwater wells at the well sites H3, H4 and H5 (Fig. 1). Depths of each site and nomenclature used for each sampling point are provided in table 1.

## 2.2 Sampling

Regular sampling was conducted monthly from May 2014 until April 2015. Water samples for $\delta^{13}C$ and $^{14}C$ analyses were obtained with a submersible groundwater pump (MP1, Grundfos, Erkrath, Germany). Samples were taken after stationary hydraulic and geochemical conditions were reached which usually required exchanging at least three well volumes. Water samples were taken following recommendations of the International Atomic Agency (IAEA, 2009). One litre Schott bottles with gastight screw caps were filled on a bypass with low water flow from bottom to top in order to avoid degassing of $CO_2$ during sampling. The bottle was rinsed two times and subsequently filled full to the brim, closed quickly with a gas tight screw cap, cooled at 4°C in the dark and transported to the lab for further analyses. For collecting material for POC and DNA, a high filtration campaign was conducted in May 2015. 1000 litres of water were pumped from well H 5.1, H 5.2 and H 4.3 through pre-combusted (500°C) glass fiber filters with 0.2 μm pore size placed on a custom filtration unit (Schwab et al., submitted). After pumping 1000 litres, filters were removed, packed into aluminium foliation and cooled on dry ice for POC and DNA analyses. To distinguish between DOC and TOC, water samples for DOC analyses were filtered through a 0.45 μm filter during sampling.

.

## 2.3 Hydrochemistry

Specific electrical conductivity, pH, water temperature, dissolved oxygen concentration and saturation were measured directly electrometrically in the field (WTW, Weilheim, Germany), using a flow through cell. Dissolved inorganic carbon (DIC), total organic carbon (TOC) and dissolved organic carbon (DOC) were determined by high temperature catalytic oxidation and a non-dispersive infrared sensor (multi 18 N/C 2100S, AnalytikJena, Germany). Nitrite, nitrate, orthophosphate, sulphate and chloride were determined by ion chromatography using an IC 20 system (Dionex, Sunnyvale,

CA) equipped with an IonPac AS11-HC column. Major cations were measured by ICP-OES (725 ES, Varian/Agilent, USA) (Kohlhepp et al., 2016).

## 2.4 $^{13}$C analyses of DIC

Stable carbon isotope ratios of DIC ($\delta^{13}C_{DIC}$) were measured according the method described by Assayag et al. (2006).
$\delta^{13}C_{DIC}$ analyses were performed on an isotope ratio mass spectrometer (IRMS) coupled to a Gasbench II (Finnigan MAT Delta$^{Plus}$ XL, Bremen, Germany) and a CTC PAL-80 autosampler. All isotope analyses were conducted within one week after sampling. In brief, 1.5 ml of water were transferred from sampled 1 litre bottles with a gastight syringe to 12 ml screw caped Labco vials with butyl rubber septa, which were pre-flushed with $N_2$. 0.5 mL of 85 % $H_3PO_4$ was added to acidify the sample to a pH below 2 and dissolved $CO_2$ was liberated by shaking and equilibrating water and headspace for 24 h. Three replicates were prepared from each water bottle for $\delta^{13}C$ analyses. Standards were prepared for linearity and drift corrections as well as normalizing measured values to the international V-PDB scale (Coplen et al., 2006).

Stable carbon isotope ratios are reported in the delta notation that expresses $^{13}C/^{12}C$ ratios as $\delta^{13}C$-values in per mil (‰) relative to the international reference material Vienna Pee Dee Belemnite (V-PDB):

$$\delta^{13}C = \left[ \frac{\left(\frac{^{13}C}{^{12}C}\right)_{sample}}{\left(\frac{^{13}C}{^{12}C}\right)_{reference}} - 1 \right] \times 1000 \tag{4}$$

Precision of $\delta^{13}C_{DIC}$ based on three repeated measurements from one sampled bottle was better than 0.3 ‰ ($\pm 1\sigma$).

## 2.5 $^{14}$C analyses of DIC

Radiocarbon concentrations of DIC were measured by accelerator mass spectrometry (AMS) at the Jena $^{14}$C facilities (Steinhof et al., 2004). Groundwater DIC was extracted by a headspace-extraction method adapted from Gao et al. (2014). In brief, 25 mL of groundwater samples, corresponding to about 1 mg C, were transferred into 60 mL I-Chem septum sealed screw cap vials within a glove bag containing an $N_2$ atmosphere. Vials were closed with Teflon/silicone septa and additionally underlain Black Viton septa (Sigma Aldrich, St. Louis, MO, USA), in order to avoid contamination of $^{14}$C depleted carbon from the septa rubber (Gao et al., 2014). After closing the vials, 0.5 ml of 85 % $H_3PO_4$ were added with a syringe to acidify the sample and convert all DIC into $CO_2$ ($CO_{2(aq)}$). The sample was shaken gently and left to equilibrate at room temperature for at least 24 hours. Three sets of standards were prepared for every batch of samples to correct for contamination either from atmosphere intrusion or from $^{14}$C free septa material. For standards containing either no or modern $^{14}$C concentrations, 17.5 mg of IAEA-C1 ( 0 pMC) and in house coral standard powder (CSTD coral, obtained from Ellen Druffel, UC Irvine, 94.45 ± 0.18 pMC) were dissolved in 25 ml acidified water, respectively. Additionally, a blank was

prepared to check for the background of the acidified water. Acidified water was prepared by adding degassed 85 % $H_3PO_4$ to ultra pure Milli-Q water (Millipore Corp., Billerica, MA, USA) until a pH of lower 2 was reached and by stripping the water with a $N_2$ stream for 1 hour.

After preparation, sample as well as standard $CO_2$ was directly extracted cryogenically from the vial headspace into a customized high vacuum extraction line using an 1:1 ethanol/dry ice mix as water trap and liquid nitrogen for freezing out the $CO_2$. Extraction efficiency was checked by measuring the pressure within the vials after $CO_2$ release and equilibration with the headspace on the extraction line.

All radiocarbon values are reported in percent modern carbon pMC, which is defined as the fractionation corrected ratio between the $^{14}C$ activity of the sample compared to the new oxalic acid standard (NOX; NBS SRM 4990C) according to (Steinhof et al., 2004):

$$pmC = R * \frac{1}{0.7459} * \left(\frac{1+\delta^{13}C_{NOX}}{1+\delta^{13}C}\right)^\theta \tag{5}$$

Errors reported for $^{14}C_{DIC}$ analyses are the external analytical precision based on repeated measurements of a control sample, which was better than 0.46 pMC ($\pm 1\sigma$).

## 2.6    $\delta^{13}C$ analyses of DOC and POC

DOC $\delta^{13}C$ values of samples from November 2014, March 2015 and May 2015 were determined on a high performance liquid chromatography (HPLC) system coupled to an IRMS (HPLC/IRMS) system (Scheibe et al., 2012). HPLC/IRMS allows direct determination of $\delta^{13}C$ values of DOC in the liquid phase by coupling a LC-IsoLink system (Thermo Electron, Bremen, Germany) to a Delta+ XP IRMS (Thermo Fisher Scientific, Germany). A detailed description of the apparatus and measurement procedure is given in Scheibe et al. (2012). Errors reported for DOC analyses represent the external analytical precision based on repeated measurements of one control sample. External precision was better than 0.15 ‰. Low DOC content coupled with high salt content of DOC prevented radiocarbon measurements on the collected samples.

$\delta^{13}C$ and $^{14}C$ values of POC were obtained by combustion of material trapped on the precombusted glass fiber filters. Pieces of filters were cut out and weighted into tin capsules. The $^{13}C/^{12}C$ isotope ratio was determined on an isotope ratio mass spectrometer (DELTA+XL, Finnigan MAT, Bremen, Germany) coupled to an elemental analyser (NA 1110, CE Instruments, Milan, Italy) via a modified ConFloII™ interface (EA-IRMS). Stable carbon isotope ratios are reported in the delta notation that expresses $^{13}C/^{12}C$ ratios as $\delta^{13}C$-values in per mil (‰) relative to the international reference material NBS 22 (equation 4). Only one sample of filter could be run for each well. Errors reported for POC values represent the precision of the analysis sequence, based on repeated analysis of a control sample.

## 2.7 DNA extraction and sequencing

DNA was extracted from the filtered groundwater using the RNA PowerSoil® Total Isolation kit followed by the RNA PowerSoil® DNA elution accessory kit (MO BIO, Carlsbad, CA, USA) following the manufacturer's protocol, and then stored at -20 °C

Groundwater DNA aliquots were shipped to LGC Genomics GmbH (Berlin, Germany) for Illumina MiSeq sequencing and the 341F - 785R primer pair was used. Because the DNA concentrations from the filter pieces were low, PCR products were used for sequencing, and not genomic DNA. This first round of PCR was carried out on DNA samples using the B8F – U1492R primer pair, and conditions for PCR were 30 cycles with 1 min at 94°C, 1 min at 55°C and 2 min at 72°C. The Illumina sequence datasets were analyzed using Mothur v.1.36.1 following the Schloss SOP (http://www.mothur.org/wiki/MiSeq_SOP). Sequences obtained in this study were deposited in the European Nucleotide Archive under accession numbers ERS1392525- ERS1392530.

## 2.8 $^{14}$C DIC age models and hydro-chemical modelling

### 2.8.1 Han-Plummer plot

In order to use DIC isotopes as a proxy for water flow and carbon turnover in the subsurface compartments of the CZ, we applied the graphical method developed by Han et al. (2012). The method is based on plotting $^{14}$C$_{DIC}$ against $^{13}$C$_{DIC}$ as well as the reciprocal of the DIC concentration. Processes affecting DIC and its isotopic composition include calcite/dolomite dissolution (Tamers et al., 1975), isotopic exchange under open or closed conditions (Fontes and Garnier, 1979; Eichinger, 1983; Han and Plummer, 2013), heterotrophic respiration of organic matter (Aravena et al., 1995) as well as precipitation and recrystallization of calcite (Wigley, 1976). Adjustment models exist for most of the mentioned processes, which can be applied in order to determine the initial $^{14}$C concentration (Han and Plummer, 2016). The Han-Plummer plot can help to identify processes that affect the DIC isotopic signature. Key features of the Han-Plummer plot are described here briefly with the aid of Fig. 6a and Fig. 6b.

On Fig. 6a, point A represents the isotope value of $CO_2$ in the recharge zone. The isotopic composition of soil $CO_2$ in the Hainich area was obtained from (Hahn, 2004), who measured soil $CO_2$ values averaging -23.00 ‰ in soils close to the recharge area. Considering a fractionation factor of -1.32 ‰ between gaseous and dissolved $CO_2$ (Mook et al., 1974), a $\delta^{13}$C value of -24.32 ‰ can be derived for dissolved soil $CO_2$ (point A'). We initially chose 100 pmC (i.e. preindustrial atmospheric $CO_2$) as the initial radiocarbon concentration of soil $CO_2$. We do not correct for mass-dependent fractionation of radiocarbon as this is accounted for in the correction of reported radiocarbon data (Trumbore et al., 2016).

Water that is equilibrated with soil $CO_2$ reacts with carbonates either under open system conditions in the soil or under closed conditions within the aquifer. Equilibration with carbonates according to equation (1) shifts DIC values in the Han-Plummer plot to the so-called Tamer´s point, which represents the $\delta^{13}C$ or $^{14}C$ value of DIC diluted by $CaCO_3$ according to:

$$^{13/14}C_0 = \left(\frac{C_a}{C_t}\right) \times {^{13/14}C_g} + 0.5\frac{C_b}{C_t}\left({^{13/14}C_g} + {^{13/14}C_s}\right) \tag{6}$$

where $^{14}C_0$ represents the $^{13}C$ or $^{14}C$ values of DIC reacted with $CaCO_3$. $C_a$, $C_b$ and $C_t$ refer to $CO_{2(aq)}$ and $HCO_3^-$ and total DIC concentrations, respectively. $^{13/14}C_g$ and $^{13/14}C_s$ refer to $\delta^{13}C$ or radiocarbon concentrations of soil gas and solid carbonate, respectively.

In our study, Tamer´s point is located at -12.16 ‰ for $\delta^{13}C$ and 50 pmC, considering a carbonate isotopic end-member of 0.29 ‰ $\delta^{13}C$ and 0 pmC (point C in Fig. 6a). Isotopic exchange, which can occur either in the soil by reaction between soil $CO_2$ and DIC or in the saturated zone between DIC and solid carbonates can also be identified with the graphical method. Isotopic exchange in the soil zone, which is usually accompanied with slow water infiltration (Han and Plummer, 2016), would shift $\delta^{13}C$ and $^{14}C$ from Tamer´s point towards values close to A´, whereas the endpoint of DIC fully equilibrated with respect to soil $CO_2$ can be derived according to (Han and Plummer, 2016):

$$\delta^{13}C_0 = \left(\frac{C_a}{C_t}\right) \times \left(\delta^{13}C_g + \varepsilon_{a/g}\right) + \left(\frac{C_b}{C_T}\right) \times \left(\delta^{13}C_g - \varepsilon_{g/b}\right) \tag{7}$$

and

$$^{14}C_0 = \left(\frac{C_a}{C_T}\right) \times \left({^{14}C_g} + 0.2\varepsilon_{a/g}\right) + \left(\frac{C_b}{C_T}\right) \times \left({^{14}C_g} - 0.2\varepsilon_{g/b}\right) \approx {^{14}C_g} \tag{8}$$

where $^{14}C_0$ and $\delta^{13}C_0$ represents the $^{13}C$ or $^{14}C$ values of DIC equilibrated with soil $CO_2$. $C_a$, $C_b$ and $C_t$ refer to $CO_{2(aq)}$ and $HCO_3^-$ and total DIC concentrations, respectively. $\delta^{13}C_g$ and $^{14}C_g$ refer to stable isotope composition and radiocarbon concentrations of soil $CO_2$, respectively. $\varepsilon_{a/g}$ and $\varepsilon_{g/b}$ are the respective carbon isotope fractionation factor of gaseous $CO_2$ and dissolved $CO_2$ and gaseous $CO_2$ and $HCO_3^-$.

Isotopic exchange under closed conditions would include isotope exchange reactions between DIC and $CaCO_3$ and shift $\delta^{13}C$ and $^{14}C$ values in direction to the calcite end-member C. Fully equilibrated DIC isotopic signatures can be derived according to (Han and Plummer, 2016):

$$5 \quad \delta^{13}C_0 = \left(\frac{C_a}{C_t}\right) \times \left(\delta^{13}C_s - \varepsilon_{s/a}\right) + \left(\frac{C_b}{C_T}\right) \times \left(\delta^{13}C_s - \varepsilon_{s/b}\right) \tag{9}$$

and

$$^{14}C_0 = \left(\frac{C_a}{C_t}\right) \times \left(^{14}C_s - 0.2\varepsilon_{s/a}\right) + \left(\frac{C_b}{C_T}\right) \times \left(^{14}C_s - 0.2\varepsilon_{s/b}\right) \tag{10}$$

where $\varepsilon_{s/a}$ and $\varepsilon_{s/b}$ refers to carbon isotope fractionation factor of $CaCO_3$ and $CO_{2(aq)}$ and $CaCO_3$ and $HCO_3^-$, respectively.

End-members for isotopic exchange under closed conditions are represented by point D and by point E in Fig. 6a. DIC that is affected by isotopic exchange to various extents, either under closed or open conditions, would shift DIC isotope values along the dashed lines between Tamer´s point and points E and D in Fig. 6a.

Other processes that might affect the isotopic composition of DIC include heterotrophic respiration of organic matter, which can be linked to a variety of microbial metabolisms (Han et al., 2012). The influence of heterotrophic respiration on DIC can be obscured, if oxidized OM in the aquifer has similar $\delta^{13}C$ or $^{14}C$ values to soil $CO_2$. However, by plotting $\delta^{13}C$ or $^{14}C$ against the reciprocal of the DIC concentration, heterotrophic respiration can be identified, because DIC usually increases in this case (Han et al., 2012) (Fig. 6b and c). Corresponding shifts in DIC isotopes are dependent on the isotopic composition of the oxidized organic matter.

### 2.8.2 NETPATH modelling

The mass-fluxes suggested by the Han-Plummer plot, were quantified with the inverse geochemical modelling program NETPATH (Plummer et al., 1994). NETPATH calculates the chemical evolution of waters along a real or hypothetical flow path between an initial and final well (El-Kadi et al., 2011). Models include inverse geochemical calculations, equations for chemical and isotopic mass balance as well as Rayleigh equations for evolutionary paths of isotopes in aquifers. The models are adjusted to measured mineralogy, as well as chemical and isotopic composition of the groundwater (El-Kadi et al., 2011). A unique feature of NETPATH is its ability to perform radiocarbon age estimates for groundwater by applying traditional $^{14}C$ adjustment models and accounting additionally for water rock interactions, mixing, redox reactions and isotope exchange processes.

For determining carbon evolution within the flow path, NETPATH uses the concept of total dissolved carbon (TDC) (Plummer et al., 1994):

$$mTDC = mDIC + mCH_4 + mDOC \tag{10}$$

NETPATH accounts also for organic matter oxidation, including its isotopic composition, which is not done by traditional [14]C adjustments models. Calculated [14]C activity of water $A_{nd}$ in the final well is adjusted for chemical reactions but not for radioactive decay. The radiocarbon age is subsequently calculated according to:

$$t(years) = \left(\frac{5730}{\ln 2}\right) * \ln\left(\frac{A_{nd}}{A}\right) \tag{11}$$

where 5730 represents the half-life of radiocarbon in years, A is the observed [14]C activity of TDC in the final well and t is travel time between initial and final well.

Fractionation factors for Rayleigh isotope effects were obtained from Mook et al. (1974). Constrains and phases for the calculations were chosen according to changes that were observed in the hydro-chemical data. Input data for the model is provided in the supplemental material.

We chose five wells, representing the three major water chemistry clusters in the two aquifer complexes at the Hainich CZE. Based on hydrogeological considerations (Kohlhepp et al., 2016), we estimated three flow paths for the evolution of water chemistry between wells for each case. Modelled flow paths for HTL were H-31 (initial well) to H-41(final well) and H-41 (initial well) to H-51 (final well). For HTU we modelled flow path H-32 (initial well) to H-42 (final well) (=HTU 1) and H-32 (initial well) to H-52 (final well) (=HTU 2). We considered therefore that no connection occurs between wells H-42/H-43 and H-52/H-53.

## 3    Results

### 3.1    Hydrochemistry

Within the time period of monitoring there was no significant change from previous reported hydro-chemical patterns (Table 1) (Herrmann et al., 2015; Küsel et al., 2016). Water chemistry reflects the limestone environment of the catchment. $Ca^{2+}$, $Mg^{2+}$, $HCO_3^-$, $CO_3^{2-}$ and $SO_4^{2-}$ are the main ions in the system (Table 1). The waters are characterised as earth alkaline bicarbonatic to bicarbonatic-sulfatic waters (Küsel et al., 2016). The footslope wells of HTU (H-42/43/52/53) are depleted in oxygen, whereas the upper/midslope HTU-wells (H-32) contain moderate concentrations in dissolved oxygen. By contrast HTL has higher values of dissolved oxygen ranging from 1.6 mg/L in H-31 to 0.23 mg/L in H-51. A substantial decrease in $Ca^{2+}$ is observed in HTU along the flow path.  In contrast, in HTL $Ca^{2+}$ concentrations increase, with a doubling of $Ca^{2+}$ at

location H-51 compared to H-31 or H-41. $Mg^{2+}$ concentrations are higher in HTU compared to HTL and highest at well H-52 and H-53. $K^+$ concentrations triple in HTU along the presumed flow path but remain constant at a low level in HTL. $Fe^{2+}$ concentrations are close to zero in both aquifer assemblages, but increase in well H-42 (HTU). $NO_3^-$ concentrations decline along the flow path to near zero in HTU and to < 10 mg/L in HTL. Sulphate concentrations in the majority of wells are close

to 100 mg/L but decrease in HTU at location H-42 and increase sharply in HTL in well H-51.

DIC concentrations in both aquifers are close to 70 mg/L but are higher in HTU in wells H-42 and H-43 and lower at location H-51 in HTL.

DOC concentrations are below 1 mg/L in HTU as well as HTL with temporal variations in both aquifers.

### 3.2    $\delta^{13}C$ and $^{14}C$ of DIC

The majority of wells have nearly constant $\delta^{13}C$ values, with a mean annual value of -11.7 $\pm$ 0.2 ‰ (Fig. 2). However, $\delta^{13}C_{DIC}$ in wells H-52 and H-53 of HTU are significantly more enriched compared to all other wells with values of -8.9 $\pm$ 0.3 ‰ and -9.5 $\pm$ 0.3 ‰, respectively.

Radiocarbon DIC results show a similar pattern. All wells show little temporal variation during the monitoring period. In HTL, $^{14}C$ concentrations decrease from well H-31 to H-41, but increase again in well H-51. Radiocarbon values of HTU

decrease continuously from 62.2$\pm$ 7.0 pMC in H-32 to 13.4 $\pm$ 0.5 pMC in H-53 (Fig. 3).

According to results from $\delta^{13}C_{DIC}$ and $^{14}C_{DIC}$ measurements, the observed wells can be divided into three groups (boxplots in lower part of Fig. 2 and 3). Group 1 comprises all oxic wells of HTL and well H-32 of HTU, which is also oxic. Group 2 and 3 describe the anoxic wells of HTU in location H-4 and H-5, respectively.

### 3.3    $\delta^{13}C$ DOC

Unlike DIC values, $\delta^{13}C$ of DOC shows no distinct spatial patterns, with all wells in both aquifer assemblages averaging -23.5 $\pm$1.6 ‰ without clear trends over the observation period (Table 2). We were not able to measure radiocarbon in bulk DOC.

### 3.4    $\delta^{13}C$ and $\Delta^{14}C$ of POC

In May 2015, $\delta^{13}C$ values for POC are most depleted in well H-43 ( -33.8 $\pm$ 0.2 ‰) compared to wells H-51 and H-52 (-28.7

$\pm$ 0.2 ‰ and -28.4 $\pm$ 0.2 ‰, respectively; Table 2). These are in turn lighter than the $\delta^{13}C$ signatures of 45-cm deep soil organic matter (SOM) in the recharge zones of the HTL and HTU (-25.8 $\pm$ 0.18 and -26.1 $\pm$ 0.2, respectively).

Radiocarbon values of POC are 74.4 $\pm$ 0.3 pmC in well H-51 (HTL) and 41.2 $\pm$ 2.0 pmC in H-43 and 35.2 $\pm$ 0.2 pMC in H-52 (both HTU). These are lower than those of SOM (45 cm depth) in the recharge zone of HTL and HTU (88.5 $\pm$ 1.4 and 87.5 $\pm$ 0.3).

Carbonates of both aquifer assemblages are free of radiocarbon (i.e. 0 pMC) and have an average $\delta^{13}C$ value of 0.3 ± 0.3 ‰.

### 3.5    Bacterial and archaeal 16S rRNA gene diversity

The bacterial community diversity in the three well samples display different patterns (Fig. 4). The sample from well H-43 (Group 2) is dominated by Proteobacteria (30.9 % of the total reads) and Candidate Division OD1 (12 %), and also composed of Firmicutes, Candidate Division OP3, Nitrospirae, Bacteroidetes and Chloroflexi. On the genus level, the most dominant groups are the sulfate-reducing *Desulfosporinus* (5.2 %) of the Firmicutes, unclassified genera of the Gallionellaceae family (2.1 %, Betaproteobacteria), and unclassified groups belonging to the Deltaproteobacteria (8.8 %).

The sample from well H-51 (Group 1) is dominated by Proteobacteria (32.4 %), and also composed of Nitrospirae and Actinobacteria. On the genus level, the most dominant groups are *Albidiferax* (8.4 %, Betaproteobacteria), *Nitrospira* (5.9 %), unclassified genera of the Caulobacteraceae family (4.1 %, Alphaproteobacteria), *Aquicella* (1.3 %) and *Acidiferrobacter* (1.2 %) of the Gammaproteobacteria, and unclassified groups belonging to the Deltaproteobacteria (5.2 %).

The sample from well H5-2 (Group 3) is dominated by Proteobacteria (28.5 %), Chlorobi (26 %) and Candidate Division TM7 (16.6 %), and is also composed of Nitrospirae and Bacteroidetes. On the genus level, the most dominant groups are *Syntrophus* (5.1 %), *Desulfovibrio* (2 %) and *Desulfocapsa* (1 %) of the Deltaproteobacteria, *Sulfuritalea* (2.6 %, Betaproteobacteria), and *Sulfuricurvum* (1 %, Epsilonproteobacteria.

The archaeal community structure also shows distinct patterns for all three groups (Fig. 5). The samples from wells H-51 (Group 1) and H-52 (Group 3) were almost solely composed of the ammonia-oxidizing Marine Group I (MG-I) Thaumarchaeota. The sample from H-52 (Group 3 well) also contained sequences affiliated with the subgroup 7/17 of the Bathyarchaeota (6.3 % of the total reads).

Sample H-43 (Group 2 well) exhibits the most diverse archaeal community. The most dominant groups were unclassified genera of the hydrogenoclastic (using $H_2$ and $CO_2$) methanogenic family *Methanoregulaceae* (22 %); subgroups -6 and -11 of the Bathyarchaeota (19.9 % and 11.5 %); and subgroup 1.1c of the *Thaumarchaeota* (12.9 %). The MG-I *Thaumarchaeota* which dominated the samples from wells H-51 and H-52 represented only 1.6 % of the archaeal community in well H-43.

### 3.6    Graphical evaluation of radiocarbon data

Concordant to results of the $\delta^{13}C_{DIC}$ and $^{14}C_{DIC}$ monitoring, three clusters of points can be distinguished when DIC data are plotted in the Han-Plummer plots (Figs. 6a-c). The first group falls on Tamer´s point for $\delta^{13}C$ but is enriched in $^{14}C$ (arrow d in Fig. 6a). This group (Group 1) comprises all wells of HTL and well H-32 of HTU.

Group 2 also falls close to Tamer´s line for $\delta^{13}C_{DIC}$, but is depleted in $^{14}C$ and has elevated DIC concentrations (arrow e in Fig. 6a and arrow b in 6b and 6c). This pattern is indicative for oxidation of organic matter that is depleted in $^{14}C$ but close to SOM $\delta^{13}C$, accompanied with calcite dissolution. Group 2 comprises wells HTU H-42 and H-43.

Wells HTU H-52 and H-53 constitute the third group (Group 3). Group 3 wells fall off Tamer´s point for $\delta^{13}C$ and $^{14}C$ towards more enriched $\delta^{13}C$ and depleted $^{14}C$ values. Enriched $\delta^{13}C$ and depleted $^{14}C$ values can be indicative of enhanced calcite dissolution (Han et al., 2012). However, both wells fall off the calcite dissolution line (arrow b in figure 6a) and they are shifted towards more depleted $\delta^{13}C$ values (arrow c in Fig. 6a), indicating the influence of more depleted organic C sources or dissolution-precipitation processes (line e in Fig. 6a).

### 3.7   NETPATH modelling

For the flowpath H-31 to H-41 (Group 1 wells) hydrochemical composition and isotope values for DIC can be reconstructed assuming mainly reactions between water and carbonate rock. Computed stable isotopes of DIC are off the $1\sigma$ uncertainty for $\delta^{13}C_{DIC}$ ($\Delta=0.72$ ‰, $p>0.01$), but reproduce measured radiocarbon values ($\Delta=0.11$ pmC, $p=0.79$) (Table 4). For calculating initial $^{14}C$ concentrations, Tamer´s model provides the best match, which is in accordance with the graphical method. Dissolution of 1.01 mmol L$^{-1}$ of calcite is required to explain the evolution of water chemistry between the two wells (Table 3).

For the flowpath H-41 to H-51 (Group 1 wells) $\delta^{13}C_{DIC}$ values can be computed assuming 0.31 mmol L$^{-1}$ calcite dissolution. Computed values are within the $1\sigma$ uncertainty of measured values ($\Delta=0.11$ ‰, $p=0.38$) (Table 4). However, there is a less good agreement for radiocarbon ($\Delta=5.83$ pmC, $p=0.07$) and DIC in the final well has higher $^{14}C$ concentrations than in the initial well. NETPATH computes modern radiocarbon ages for both flow paths in HTL, i.e. water was recharged recently and travel time between initial and final well cannot be resolved by radiocarbon dating.

Two separate flow paths were assumed for modelling the evolution of water chemistry in the upper aquifer assemblage (HTU). The best match for flow path HTU-1 (H-32 to H-43) was found by oxidizing 0.8 mmol L$^{-1}$ of organic matter coupled to iron reduction (dissolution of 0.18 mmol L$^{-1}$ goethite) and sulphate reduction (precipitation of 0.18 mmol L$^{-1}$ of pyrite) (Table 3).  The best match for calculating isotope values was obtained using the values measured in POC as the source of oxidized C (Table 4). Calculated isotope values using the model match well for $\delta^{13}C_{DIC}$ ($\Delta=0.01$ ‰, $p=0.91$) but less well for $^{14}C_{DIC}$ ($\Delta=2.92$ pmC, $p=0.07$) (Table 4). Water travel time between H-32 and H-43 is computed as 403 years.

The complexity of reactions increases in flowpath HTU-2. No match could be found by using Tamer´s model for $A_0$ determination. The revised Fontes & Garnier model for closed system isotope exchange was used instead. Reactions necessary for the inverse model include calcite dissolution and isotopic exchange (2 mmol L$^{-1}$), dolomite dissolution, sulphate reduction and methanogenesis as well as removal of ammonia (Table 1). In order to keep the system in balance 3.78 mmol L$^{-1}$ of $CO_2$ have to be removed from the system. Calculated radiocarbon values match measured values ($\Delta=0.96$ pmC,

p= 0.002) as well as $\delta^{13}C_{DIC}$ ($\Delta$=0.57 ‰, p= 0.007) (Table 4). Water travel time between H-32 and H-52 is computed as 295 years.

Because this model suggests the formation of high amounts of methane, which was not observed in the aquifer (T. Behrendt, unpublished data), we ran a second model including more depleted values for POC. These values were based on PLFA data from (Schwab et al., unpublished), who measured $^{14}C$ activity and $\delta^{13}C$ of the main bacterial biomarker C16:0. Measured vales were -39.83 ± 0.72 ‰ for $\delta^{13}C$ and 7 ± 1 pMC for radiocarbon. Assuming this C source, model-estimated $^{14}C$ and $\delta^{13}C$ values agree less well with observations ($\Delta$=3.69 pMC, p= < 0.01 and $\Delta$=0.67 ‰, p= < 0.01 for $^{14}C_{DIC}$ and $\delta^{13}C_{DIC}$, respectively), but no methane production is required by the model. Water travel time in this case is computed as 587 years.

## 4    Discussion

### 4.1    Groundwater flow

DIC isotopes for the three different groundwater groups reflect both, differences in recharge characteristics as well as the evolution of waters as they flow through the aquifers.

The carbon isotopic composition of DIC of Group 1 waters suggests rapid recharge and minor exchange reactions due to rapid flow. $\delta^{13}C$ values are close to Tamer´s point, typical for fast water infiltration without isotopic exchange between DIC and $CO_2$ or carbonates in the soil (Han and Plummer, 2013). However, $^{14}C$ values of Group 1 wells are enriched compared to the preindustrial atmosphere assumed for Tamer's point soil $CO_2$ end-member (i.e. $^{14}C$ signature of 100 pmC for coil $CO_2$). Enrichment of $^{14}C$ can be caused by isotope exchange between soil $CO_2$ and DIC in the unsaturated zone (Fontes and Garnier, 1979; Han and Plummer, 2013). However, measured DIC values do not fall on line g in figure 6a, which would be indicative for isotope exchange under open conditions in the soil. Enrichment of $^{14}C$ along arrow d in figure 6a indicates rather the presence of excess $^{14}C$ derived from nuclear bomb testing (Gillon et al., 2012).  The presence of bomb carbon infers that soil $CO_2$ in the recharge zone is mainly derived from root respiration and mineralized organic matter in the topsoil (Richter et al., 1999). This is a reasonable assumption, because the soil groups in the recharge area of HTL are mainly rather shallow, belonging to Rendzic Leptosol or Cambisol soils, with presumably fast infiltration (Kohlhepp et al., 2016).

Radiocarbon dating of modern groundwater is difficult due to high model input uncertainty (Gillon et al., 2012). Recalculating initial $^{14}C$ activities of Group 1 wells assuming a Tamer´s like dilution by $^{14}C$ dead carbonates yields values of 110 pmC, 99 pmC, 120 pmC and 136 pmC for well H-31, H-41, H-51 and H-32, respectively. Initial $^{14}C$ values indicating predominance of bomb-derived radiocarbon as high as 136 pmC could be interpreted to suggest the time from the groundwater recharge area to the groundwater well in the range of years to decades. However, intense fracturing and karstification in the HTL aquifer with broad fractures, secondary porosity and even karst breccia within- and up to 4km away from the capture area (Kohlhepp et al., 2016), suggest that groundwater flow is much more rapid.  The radiocarbon

signatures thus most likely indicate variations of $^{14}C$ in soil $CO_2$ in the recharge area. For well H-51, there is also indication that $^{14}C$ values are influenced by mixing of waters of different radiocarbon concentration (Cartwright et al., 2012; Bethke and Johnson, 2008). This is supported by the increase of $SO_4^{2-}$ and $Ca^{2+}$ concentrations in well H-51, caused most probably by mixing with sulphate-rich waters from the aquifer system below HTL. Mixing of waters and associated mixing corrosion can also explain the calcite dissolution suggested by NETPATH.

Group 2 waters representing wells H-42 and H-43 in the HTU aquifer complex also fall on the Tamer´s line for $\delta^{13}C$ but not for $^{14}C$. $^{14}C$ values are more depleted than would be expected for a Tamer´s-like dilution. Tamer´s model is also proposed by NETPATH to estimate initial $^{14}C$ in the recharge area, which – because it is lower (averaging $45.51 \pm 3.62$ pmC, equivalent to 403 $^{14}C$ years) than the current atmospheric $^{14}CO_2$ values in the year of sampling (~103 pmC) – indicates the influence of $CO_2$ derived from decomposition of older organic matter. As for Group 1 wells, the hydrology in this hillslope indicates rapid movement of water (Kohlhepp et al., 2016), so depleted $^{13}C$ values in Group 2 wells do probably not represent lower flow dynamics and radioactive decay, but turnover of $^{14}C$ depleted carbon in Group 2 wells.

Isotope values of Group 3 cluster in a very different region than Groups 1 and 2. Enrichment in $^{13}C$ and depletion in $^{14}C$ is indicative of calcite dissolution. The best match between calculated and measured values was obtained using the revised Fontes & Garnier model for isotope exchange between DIC and carbonates to correct for initial $^{14}C$ in the starting well H-32 (Han and Plummer, 2013). This indicates different recharge patterns or an alternative recharge area as well as enhanced water-rock interactions in this portion of the HTU aquifer complex.

An alternative recharge area for Group 3 wells is probably located in the agricultural area at the foot slope of the sub-catchment, where H-5 wells are situated. This would imply penetration of surface waters through more than 50 meters of low-permeable cap rocks (claystones, marlstones; Erfurt formation and Warburg formation) although this is rather unlikely (Kohlhepp et al., 2016).

Group 2 and Group 3 wells have inferred radiocarbon ages of 403 and 296 or 587 years, respectively, and are therefore not regarded as modern by NETPATH. Nevertheless, computed ages cannot be distinguished from modern waters within the model uncertainty.

## 4.2 Biogeochemical processes affecting DIC in Group 2 wells

As indicated by the graphical method, DIC in Group 2 wells is influenced by oxidation of organic matter. However, that OM oxidation has to be performed under anoxic conditions present in these waters. According to the NETPATH model, OM oxidation can be linked to iron and sulphate reduction. Both processes can also result in dissolution of carbonates and add OM-derived DIC according to:

$$C_6H_{12}O_6 + 24Fe^{3+} + 12H_2O \rightarrow 6HCO_3^- + 24Fe^{2+} + 30H^+ \tag{12}$$

and

$$CH_2O + SO_4^{2-} \rightarrow H_2S + 2HCO_3^- \tag{13}$$

$$H_2S + CaCO_3 \rightarrow HCO_3^- + Ca^{2+} + HS^- \tag{14}$$

A good correlation between sulphate concentrations and $\delta^{13}C$ of DIC supports an influence of sulphate reduction on DIC (Fig. 7). Moreover, a high fraction of *Desulfosporinus* species, endospore-forming strictly anaerobic sulfate-reducing bacteria (Stackebrandt et al., 1997), was detected in H-43 (Group 2 well).

With the NETPATH model, the addition of oxidized OM with isotopic signatures identical to measured POC matched observed DIC isotopic values reasonably well (Table 4). This suggests that the carbon source is depleted in $\delta^{13}C$, with $^{14}C$ close to DIC values, i.e. older organic matter.

Autotrophic organisms can provide carbon that is characterized by the above-mentioned features. Chemolithoautotrophic microorganisms have been recognized to be important components of aquifer foodwebs, if sufficient electron donors are available (Hutchins et al., 2016; Alfreider et al., 2012; Kellermann et al., 2012). Chemolithoautotrophs can use six different metabolic pathways to fix DIC, of which the Calvin-Benson Cycle and the acetyl-CoA pathway are the two most important ones (Fuchs, 2011). Herrmann et al. (2015) found that up to 17 % of the microbial community in well H-43 has the potential for chemolithoautotrophic $CO_2$ fixation via the Calvin-Benson Cycle linked to the oxidation of reduced sulphur and nitrogen compounds. The Calvin Benson Cycle can function under anoxic and oxic conditions, whereas microorganisms using the acetyl-CoA pathway are restricted to strictly anoxic conditions (Fuchs, 2011;Berg, 2011). Both pathways fractionate against $^{13}C$ and may generate carbon that has $^{14}C$ values close to DIC. Furthermore, autotrophic groups which oxidize iron like the microaerophilic Gallionellaceae (Emerson et al., 2010)) were identified in Group 2 wells ; as well as autotrophic hydrogenoclastic methanogens.

According to mass fluxes calculated with NETPATH (0.77 mmol L$^{-1}$) C derived from oxidation of $^{13}C$ and $^{14}C$ depleted organic carbon would constitute 11 % of the DIC pool.

### 4.3 Biogeochemical processes affecting Group 3 wells

The Han-Plummer plot suggests that DIC in Group 3 wells is influenced by water rock interactions, including calcite and dolomite dissolution as well as isotopic exchange between DIC and calcite (Wigley, 1976;Han and Plummer, 2016).

Quantification of these fluxes by NETPATH estimates 1.57 mmol L$^{-1}$ of calcite and 0.18 mmol L$^{-1}$ of dolomite dissolution, with exchange of 2 mmol L$^{-1}$ of carbon between DIC and carbonate rock.

Calcite dissolution/precipitation can be triggered by cation exchange of $Ca^{2+}$ with $Na^+$ and $K^+$ or $NH_4^+$ on clay minerals or organic matter (Coetsiers and Walraevens, 2009; van Breukelen et al., 2004). Decreasing $Ca^{2+}$ concentrations can cause

changes in $CaCO_3$ saturation indices and result in enhanced calcite dissolution. The impact of cation exchange on DIC isotopic values in Group 3 wells is supported by good correlations with $K^+$, $Na^+$ and $NH_4^+$ concentrations in HTU (Fig. 7). Ammonia and K input into Group 3 wells could also be an indicator for surface derived nutrients from agricultural fields and highlight the potential impact of land use change on apparent groundwater ages. These also support the idea that there is vertical transport from the agricultural zone into this part of the HTU aquifer assemblage.

In addition to abiotic exchanges with carbonate rocks, biotic processes also influence isotopic signatures of DIC in Group 3 wells. This is suggested by the Han-Plummer plot, as $\delta^{13}C$ values are more depleted than would be expected for solely isotopic exchange (Fig. 6a). NETPATH modelling suggests either involvement of methane or oxidation of organic matter that is depleted in both $\delta^{13}C$ and $^{14}C$ compared to water equilibrated with surface soil $CO_2$ (Table 3).

Depending on isotopic input parameters, NETPATH indicates that either 1.33 mmol $L^{-1}$ of methane or 1.82 mmol $L^{-1}$ of $^{14}C$ depleted OM have to be oxidized to explain DIC isotopic signatures. However methane concentrations in Group 2 and 3 wells are low compared to what has been observed in other methanogenic aquifers, with values of ~ 0.1 µmol $L^{-1}$ (T. Behrendt, unpublished data). Hence, either there is a very rapid microbial turnover of methane, which would result in low *in situ* concentrations, or oxidation of $^{13}C$ and $^{14}C$ depleted organic carbon seems a more plausible explanation, one that is further supported by the molecular data. Indeed, bacteria of the family Ignavibacteriaceae, a major group in Group 3 wells, are chemoorganotrophs growing on carbohydrate fermentation (Iino et al., 2010).

A major carbon source that might provide $^{14}C$ depleted OM can be sedimentary organic matter (Aravena and Wassenaar, 1993; Coetsiers and Walraevens, 2009). Sedimentary organic matter can be released by carbonate dissolution, or be derived from marlstones that are interbedded with the fractured carbonate rocks (Kohlhepp et al., 2016). Bathyarchaeota, which are the second abundant archaeal group in Group 3 wells, are especially metabolically versatile and able to use different carbon sources (Lazar et al., 2016b).

$^{13}C$ and $^{14}C$ depleted C for heterotrophic growth can also be (as in Group 2 wells) derived from chemolithoautotrophic microorganisms (Hutchins et al., 2016). Facultative chemolithoautotrophic bacteria like ammonia-oxidizing Nitrospirae or the thiosulfate-reducing *Sulfuritalea* (Kojima and Fukui, 2011) and the sulfur-oxidizing *Sulfuricurvum* (Kodama and Watanabe, 2004) were detected in well H-52; as well as chemolithoauotrophic ammonia-oxidizing Thaumarcheota (Berg et al., 2014) which constitute 90 % of total archaeal reads.

An additional source of $^{14}C$ depleted carbon could also be input of old C from the soil. The NETPATH models predict distinct recharge areas for group 3 wells compared to group 2 and 1, and soils in areas recharging Group 3 wells may also have slower water infiltration (Küsel et al., 2016). During slow percolation of water through the soil, extensive recycling (either through sorption-desorption or fixation and oxidation in microbial C) might lead to input of DOC from subsoil horizons that is much more depleted in $^{14}C$ than in the shallow soils of the recharge area of Group 1 and 2 wells (Schiff et al.,

1997). Also anthropogenic influences like land use change can lead to mobilization of old, [14]C depleted carbon from soils (Kalbitz et al., 2000).

Overall, while there are a range of possible sources for [13]C and [14]C depleted carbon that might contribute to the observed $^{13}C_{DIC}$ shift, it is clear that old organic matter is being oxidized to support a significant fraction of the microbial food web in Group 3 waters.

Variable carbon utilization by microorganisms has been shown already for deep surface communities (Simkus et al., 2016). Our data similarly indicate a variable substrate utilization in shallow aquifers.

The timescales, on which biotic and abiotic processes act that influence DIC isotopic composition, remains uncertain. Radiocarbon ages calculated by the models have high uncertainties and [14]C signatures inferred by models could represent variations in the signatures of [14]C in recharge zones, i.e. through differences in the degree of interaction with older C found in deeper soil layers, rather than the time it takes for water to transport C. Thus it is not possible to distinguish Group 3 water from modern waters. Water rock interactions like calcite precipitation or dissolution have been shown to act very rapidly (Miller et al., 2016; Matter et al., 2016), which is also supported by our data. Without independent information (e.g. from hydrogeologic modelling) on water fluxes through the aquifers, we cannot estimate rates of transformation associated with changes in water chemistry between wells.

According to the NETPATH model, 1.82 mmol L$^{-1}$ of OM were oxidized in Group 3 wells, which would account for more than 28 % of the total DIC pool.

## 5    Conclusions

The evaluation of DIC carbon isotopes by graphical and numerical methods revealed strikingly different abiotic and biotic processes influencing the DIC isotopic composition of groundwater in our detailed study. The combination of the Han-Plummer plot and geochemical modelling yielded consistent results and allowed us to identify and quantify the different processes contributing to these large variations in DIC within a single sub-catchment.

As the residence time of groundwater in the aquifer assemblages is expected to be rather short, we attribute the observed differences in DIC in variations in recharge areas and subsequent biogeochemical processes. In HTL wells, the recharge water (largely recharging in upslope forest) was equilibrated with $CO_2$ dominated by post-bomb C fixed from the atmosphere in the last years to decades. Subsequent evolution was largely dominated by abiotic processes, including carbonate dissolution and mixing with other waters.

In HTU wells, DIC isotopes reflected oxidation of organic matter from different sources in addition to abiotic water-rock interactions. Microbial communities in HTU wells are capable of using a range of C sources with very old [14]C concentrations (i.e. <50pmC), including inorganic carbon fixed through chemolithoautotrophy and/or heterotrophic oxidation of

sedimentary organic matter. This metabolic versatility is also supported by bacterial and archeal DNA data. As both aquifers studied are shallow and situated within a dynamic fractured setting, both biotic and abiotic processes appear to act on short timescales.

# 6 Acknowledgements

We thank Heiko Minkmar and Robert Lehmann for help with water sampling and on site logistics. We kindly acknowledge Petra Linke, Axel Steinhof and Heike Machts for measuring $\delta^{13}C$ and $^{14}C$ samples. Further, we would like to thank Christian Seifert and Sandy Laschke for assistance in sample preparation for isotope analyses. We also kindly acknowledge Xiaomei Xu from the University of California Irvine for introduction in radiocarbon measurements of DIC. The study has been conducted in the B03 project of the Collaborative Research Center 1076 AquaDiva. The work has been funded by the German Research Foundation (Deutsche Forschungsgemeinschaft, DFG) CRC 1076 "AquaDiva". MN was supported by the DFG-funded research training group GRK1257 'Alteration and Element Mobility at the Microbe-Mineral Interface' and the International Max-Planck Research School for Global Biogeochemical Cycles (IMPRS gBGC).

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

| Aquifer | Well | Depth | TIC [mmol L⁻¹] | DOC [mmol L⁻¹] | TOC [mmol L⁻¹] | pH | O₂ [mmol L⁻¹] | Ca²⁺ [mmol L⁻¹] | Mg²⁺ [mmol L⁻¹] | Feₜ [mmol L-1] | NH₄⁺ [mmol L-1] | K⁺ [mmol L⁻¹] | SO₄²⁻ [mmol L⁻¹] | NO₃⁻ [μmol L⁻¹] |
|---|---|---|---|---|---|---|---|---|---|---|---|---|---|---|
| HTL | H-31 | 43 m | 6.16 ± 0.0 | 0.27 ± 0.06 | 0.30 ± 0.07 | 7.25 | 0.28 ± 0.10 | 2.42 ± 0.06 | 1.94 ± 0.06 | 0.38 ± 0.44 | 4.98 ± 3.19 | 0.08 ± 0.003 | 0.72 ± 0.02 | 0.55 ± 0.13 |
| HTL | H-41 | 45 m | 6.97 ± 0.36 | 0.18 ± 0.08 | 0.19 ± 0.08 | 7.27 | 0.31 ± 0.06 | 2.66 ± 0.19 | 1.69 ± 0.12 | 2.49 ± 4.00 | 11.95 ± 4.64 | 0.13 ± 0.03 | 1.00 ± 0.23 | 0.22 ± 0.08 |
| HTL | H-51 | 84 m | 6.31 ± 0.22 | 0.18 ± 0.08 | 0.20 ± 0.09 | 7.18 | 0.18 ± 0.24 | 4.56 ± 0.11 | 1.71 ± 0.10 | 0.24 ± 0.30 | 3.74 ± 2.86 | 0.05 ± 0.006 | 3.03 ± 0.19 | 0.15 ± 0.09 |
| HTU | H-32 | 15 m | 6.70 ± 0.53 | 0.20 ± 0.10 | 0.22 ± 0.10 | 7.32 | 0.12 ± 0.03 | 2.37 ± 0.06 | 2.01 ± 0.03 | 0.07 ± 0.02 | 1.25 ± 1.96 | 0.07 ± 0.003 | 0.76 ± 0.04 | 0.55 ± 0.08 |
| HTU | H-42 | 9 m | 7.84 ± 0.43 | 0.20 ± 0.08 | 0.23 ± 0.10 | 7.16 | 0 | 2.08 ± 0.03 | 2.04 ± 0.02 | 3.14 ± 0.45 | 21.17 ± 1.96 | 0.17 ± 0.005 | 0.37 ± 0.014 | 0.004 ± 0.002 |
| HTU | H-43 | 8.5 m | 7.80 ± 0.18 | 0.21 ± 0.09 | 0.23 ± 0.08 | 7.17 | 0 | 2.21 ± 0.12 | 2.03 ± 0.03 | 1.66 ± 2.24 | 14.23 ± 4.36 | 0.14 ± 0.004 | 0.40 ± 0.017 | 0.003 ± 0.001 |
| HTU | H-52 | 65 m | 6.30 ± 0.30 | 0.19 ± 0.10 | 0.23 ± 0.10 | 7.31 | 0 | 1.71 ± 0.02 | 2.19 ± 0.04 | 1.70 ± 2.79 | 52.30 ± 3.92 | 0.25 ± 0.004 | 0.97 ± 0.05 | 0.04 ± 0.06 |
| HTU | H-53 | 47 m | 6.67 ± 0.56 | 0.18 ± 0.10 | 0.21 ± 0.10 | 7.36 | 0 | 1.50 ± 0.02 | 2.21 ± 0.03 | 0.55 ± 0.54 | 83.43 ± 18.01 | 0.36 ± 0.004 | 0.68 ± 0.03 | 0.01 ± 0.02 |

**Table 1: Hydrochemical data**

| Aquifer | Well | $\delta^{13}C_{DIC}$ [‰ V-PDB] | $^{14}C_{DIC}$ [pMC] | $\delta^{13}C_{DOC}$ [‰ V-PDB] | $\delta^{13}C_{POC}$ [‰ V-PDB] | $^{14}C_{POC}$ [pMC] |
|---|---|---|---|---|---|---|
| HTL | H-31 | -12.0 ± 0.2 | 58.1 ± 0.4 | -23.5 ± 1.2 | - | - |
| HTL | H-41 | -11.6 ± 0.2 | 50.5 ± 3.3 | -22.3 ± 2.0 | - | - |
| HTL | H-51 | -11.6 ± 0.3 | 62.2 ± 7.0 | -22.4 ± 1.7 | -28.7 ± 0.2 | 74.4 ± 0.3 |
| HTU | H-32 | -11.6 ± 0.4 | 65.3 ± 4.0 | - | - | - |
| HTU | H-42 | -11.8 ± 0.3 | 45.5 ± 3.6 | -22.9 ± 1.6 | - | - |
| HTU | H-43 | -11.4 ± 0.2 | 45.6 ± 1.2 | -25.4 ± 0.7 | -33.8 ± 0.2 | 41.2 ± 2.0 |
| HTU | H-52 | -8.9 ± 0.4 | 25.1 ± 0.7 | -22.5 ± 0.8 | -28.4 ± 0.2 | 35.2 ± 0.2 |
| HTU | H-53 | -9.5 ± 0.3 | 13.4 ± 0.5 | -23.2 ± 1.5 | - | - |

**Table 2: Measured isotopic data of DIC, DOC and POC**

| Flowpath | Calcite [iso-ex] | Dolomite | $CH_2O$ | $CH_4$ | $CO_2$ | Gyspum | Pyrite | Goethite | $NH_3$ Gas | Exchange |
|---|---|---|---|---|---|---|---|---|---|---|
| | Reaction [mmol L$^{-1}$] | | | | | | | | | |
| H-31 - H-41 | 1.01 | - | - | - | - | - | - | - | -1.46 | - |
| H-41 - H-51 | 0.31 | - | - | - | - | - | - | - | -1.78 | - |
| H-32 - H-43 | 1.53 | - | 0.77 | - | - | - | -0.18 | 0.18 | - | 1.70 |
| H-32- H-52 a | 1.57 [2.00] | 0.18 | - | 1.33 | -3.78 | -3.53 | 1.87 | - | - | 4.62 |
| H-32- H-52 b | 1.26 [2.00] | 1.18 | 1.82 | - | -3.96 | - | 0.10 | - | -3.22 | 4.10 |

Table 3: Mass fluxes derived from the NETPATH model. The unit of all displayed mass transfers is mmol L$^{-1}$. Negative leading signs indicate that the respective phase is removed from the water phase. "Exchange" refers to cation exchange and numbers in brackets in the "calcite" column refer to isotopic exchange between DIC and calcite minerals. $CH_2O$ refers to organic matter.

| Isotope values | Measured $\delta^{13}C_{DIC [‰ V-PDB]}$ ($\pm 1\sigma$) | Computed $\delta^{13}C_{DIC}$ [‰ V-PDB] | Measured $^{14}C_{DIC [pMC]}$ ($\pm 1\sigma$) | Computed $^{14}C_{DIC [pMC]}$ | Used model for $A_0$ | Input TOC $\delta^{13}C$ [‰ V-PDB] | Input TOC $^{14}C$ | Age (years) |
|---|---|---|---|---|---|---|---|---|
| H-41 | -11.6 ± 0.2 | -10.91 | 50.5 ± 3.3 | 50.9 | Tamers | -28.7 ± 0.2 | 74.4 ± 0.3 | modern |
| H-51 | -11.6 ± 0.3 | -11.74 | 62.2 ± 7.0 | 56.4 | Tamers | -28.7 ± 0.2 | 74.4 ± 0.3 | modern |
| H-42 | -11.8 ± 0.3 | -11.75 | 45.5 ± 3.6 | 48.4 | Tamers | -33.8 ± 0.2 | 41.2 ± 2.0 | 403 |
| H-52a | -8.9 ± 0.4 | -9.51 | 25.1 ± 0.7 | 27.4 | Rev. F&G | -28.4 ± 0.2 | 35.2 ± 0.2 | 296 |
| H-52b | -8.9 ± 0.4 | -9.61 | 25.1 ± 0.7 | 28.3 | Rev. F&G | -39.8 ±0.7 | 7.9 ± 2.0 | 587 |

**Table 4: Measured and with NETPATH computed isotopic data.**

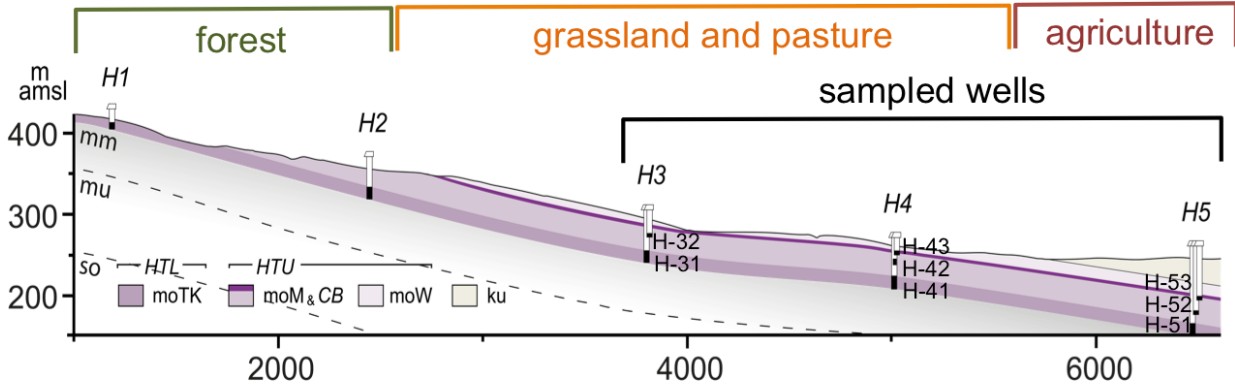

**Figure 1: Cross-section of the studied Hainich transect. Stratigraphic units moTK, moM as well as moW represent Middle Triassic units of the Upper Muschelkalk Formation. ku describes Upper Triassic sediments of the Keuper Formation. Sampled wells for this study comprised locations H3, H4 and H5. The lower aquifer assemblage HTL is recharged in the forested area in the upper part of the Hainich mountain range. Aquifer assemblage HTU is recharged in forest, grassland and agricultural areas.**

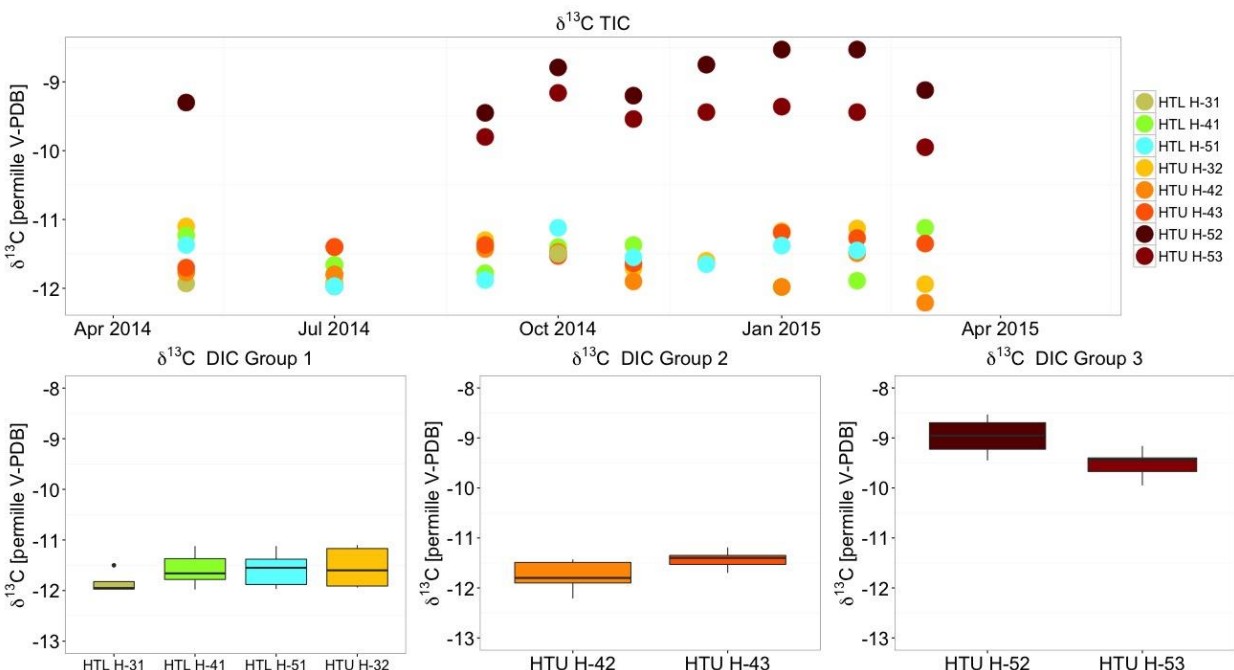

**Figure 2: Stable isotope monitoring of $\delta^{13}C_{DIC}$. Upper part: $\delta^{13}C_{DIC}$ values show minor temporal but strong spatial variations. According to results from $\delta^{13}C_{DIC}$ and $^{14}C_{DIC}$ and DIC concentrations, wells can be divided into three groups (boxplots in lower part). Group 1 comprises all oxic wells of HTL and well H-32 of HTU, which is also oxic. Group 2 and 3 are the anoxic wells of HTU in location H-4 and H-5, respectively.**

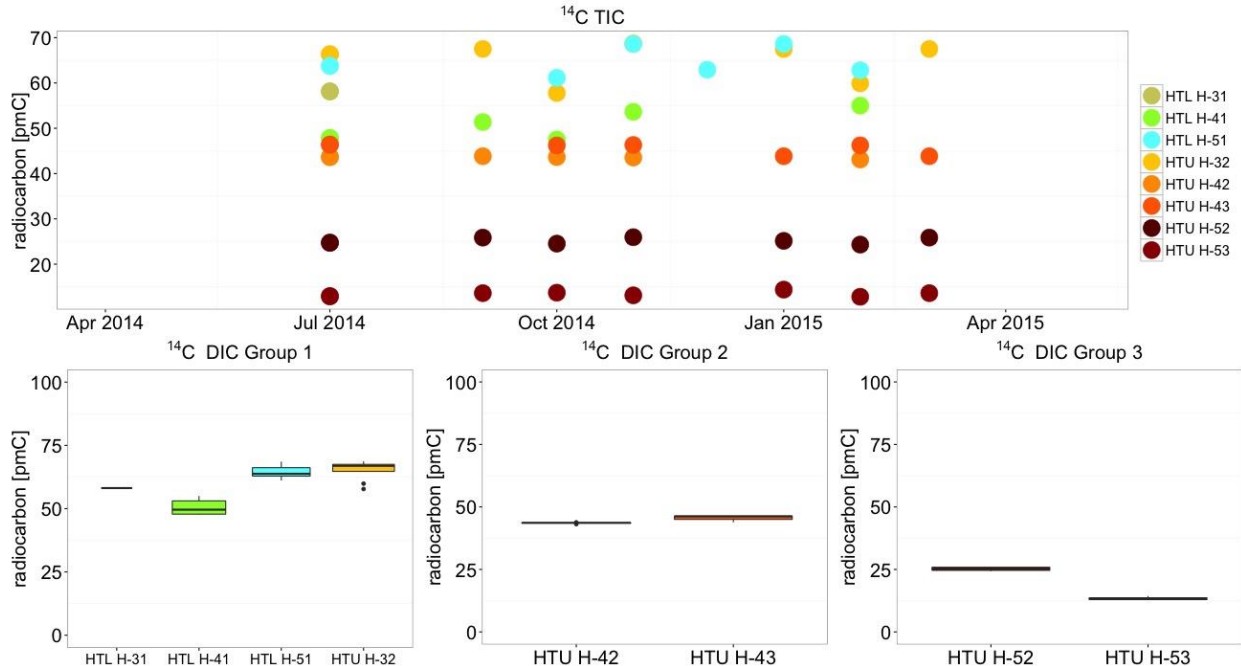

**Figure 3: Results of radiocarbon monitoring of DIC in the investigated wells. Similar to δ¹³C values, ¹⁴C_DIC values show little temporal but strong spatial variations. ¹⁴C_DIC concentrations decrease according to group 1 > group 2 > group 3. Wells of group 1 show more variation during the monitoring period than group 2 and 3 wells.**

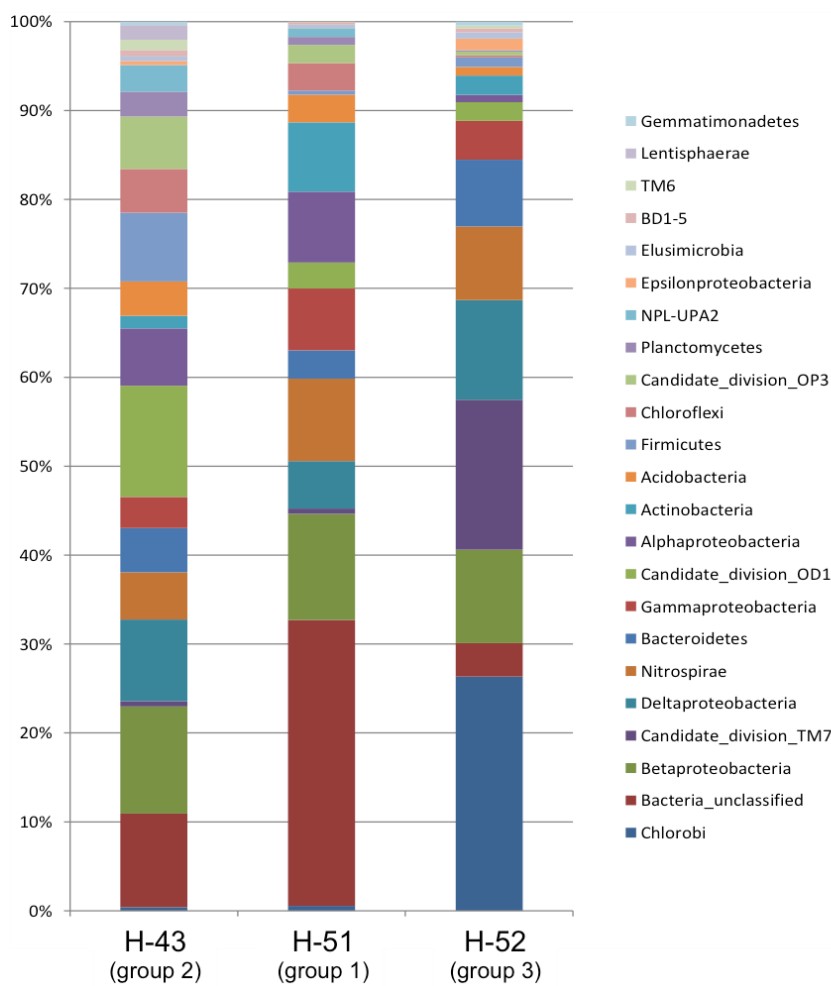

**Figure 4: Phylogenetic affiliations of DNA-based bacterial 16S rRNA gene reads in percent of total reads in groundwater samples of wells H-43, H-51 and H-52 for time point May 2015.**

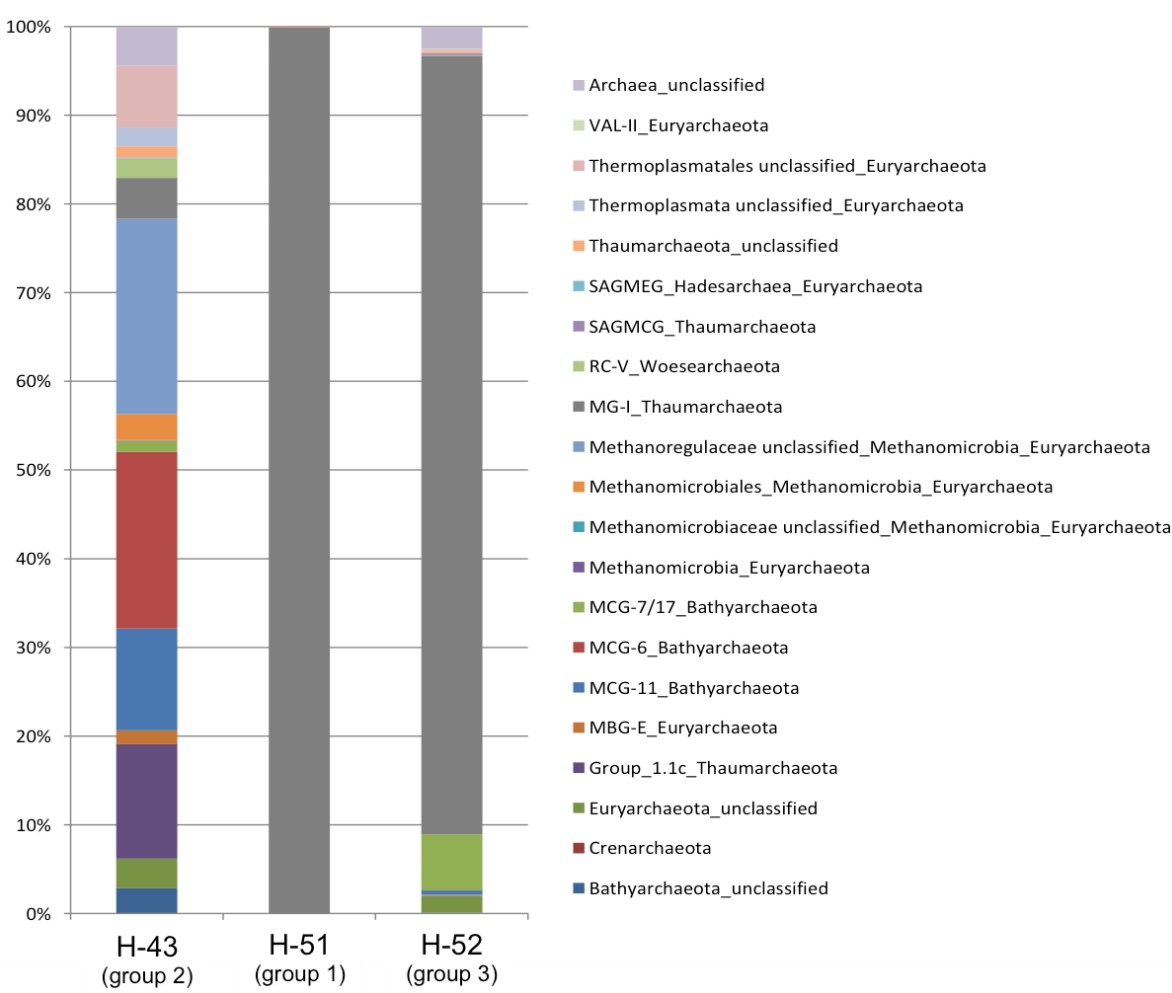

**Figure 5: Phylogenetic affiliations of DNA-based archaeal 16S rRNA gene reads in percent of total reads, for groundwater samples of all wells for time point May 2015.**

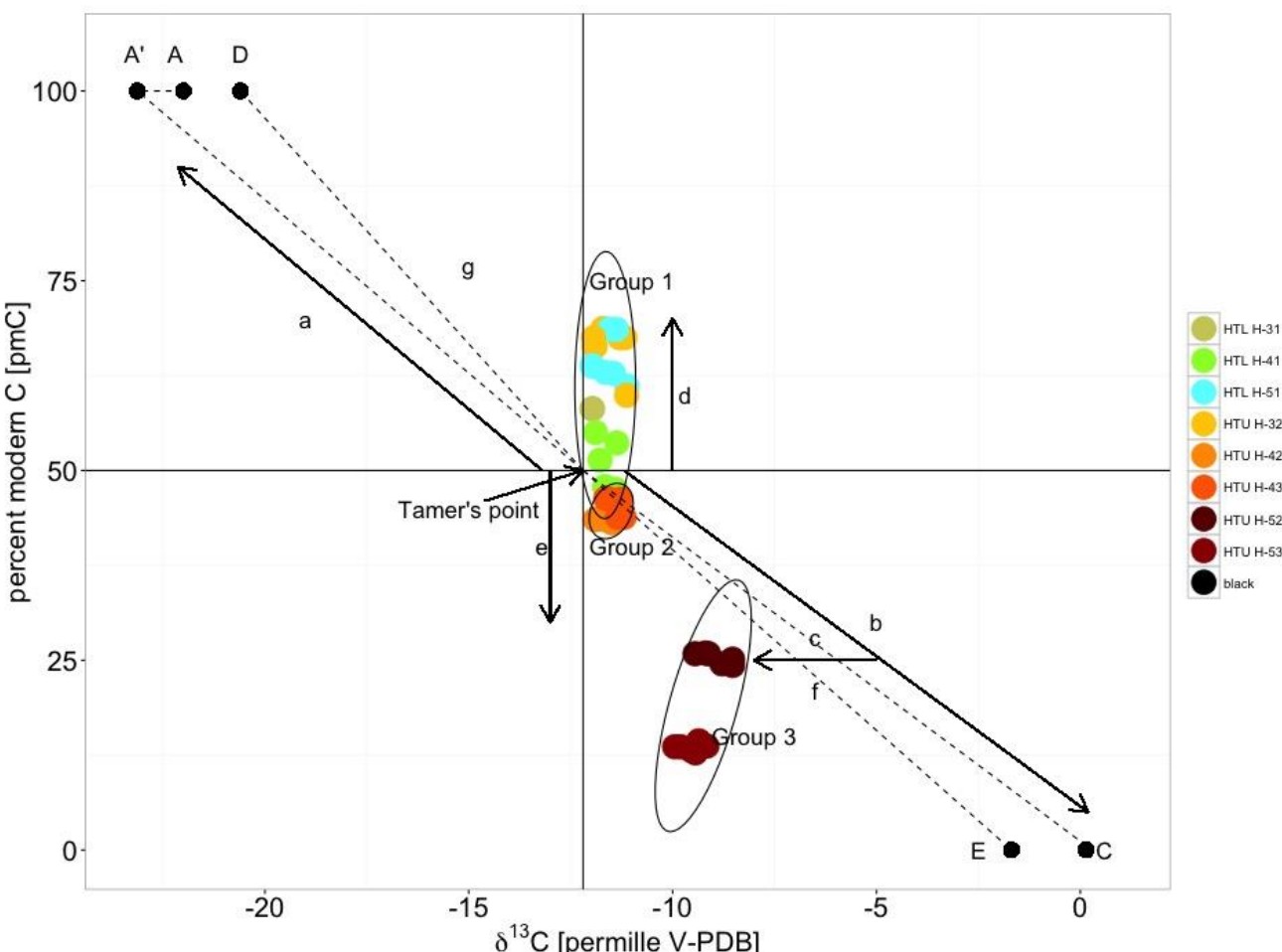

**Figure 6: Han - Plummer plot with data from groundwater sampling wells. The Methods section describes in detail the theory behind these plots and identifies the various elements shown on the figure. By plotting [14]C vs. δ[13]C three different groups can be distinguished. The oxic wells of HTL including well H-32 forming Group 1, wells H-42 and H-43 (Group 2) as well as H-52 and H-53 (Group 3). The Han-Plummer plot indicates [14]C enrichment due to bomb carbon for group 1 wells (arrow d), oxidation of [14]C depleted OM accompanied with calcite dissolution in group 2 wells (arrow e) and complex water rock interactions and OM turnover in group 3 wells (arrow c). Further explanations are given in the text.**

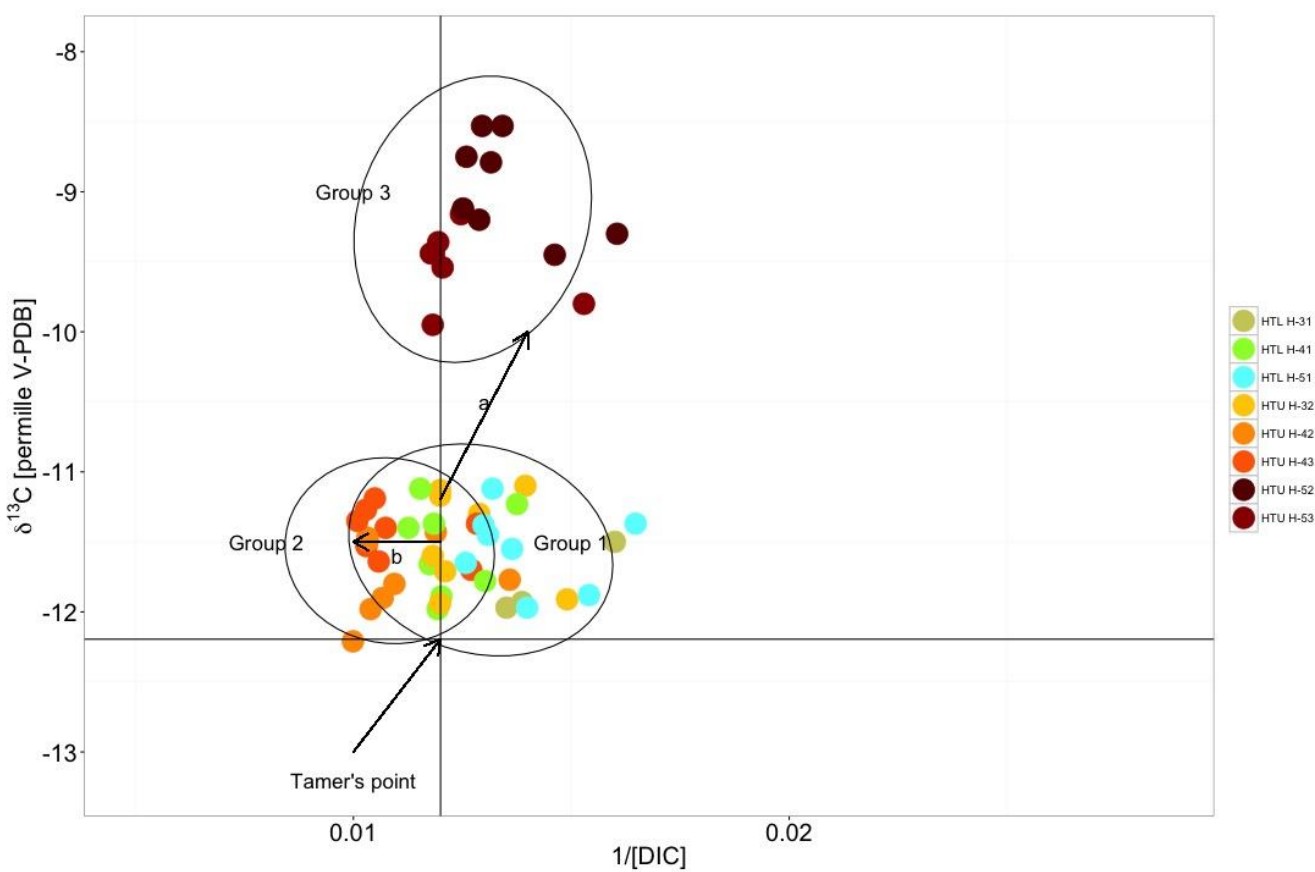

**Figure 6b: Plotting $^{13}$C vs. 1/DIC [mmol L$^{-1}$] also allows distinguishing the three groups. Group 2 is not distinct in $^{14}$C compared to Group 1, but differs in DIC concentration. Group 2 is distinct from group 1 in $\delta^{13}$C but does not differ in DIC concentration.**

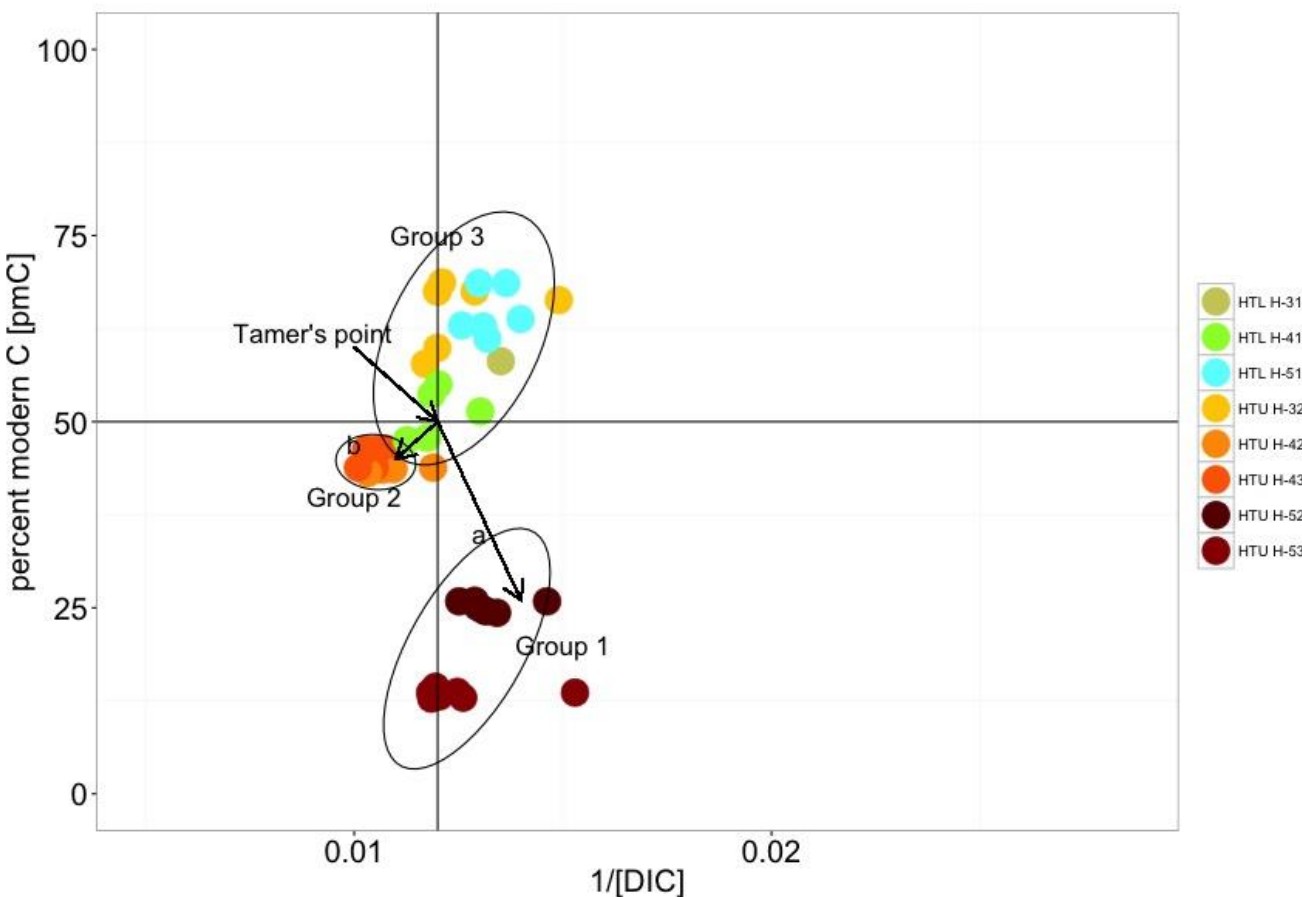

**Figure 6c: Plotting δ¹³C concentrations against 1/[DIC].**

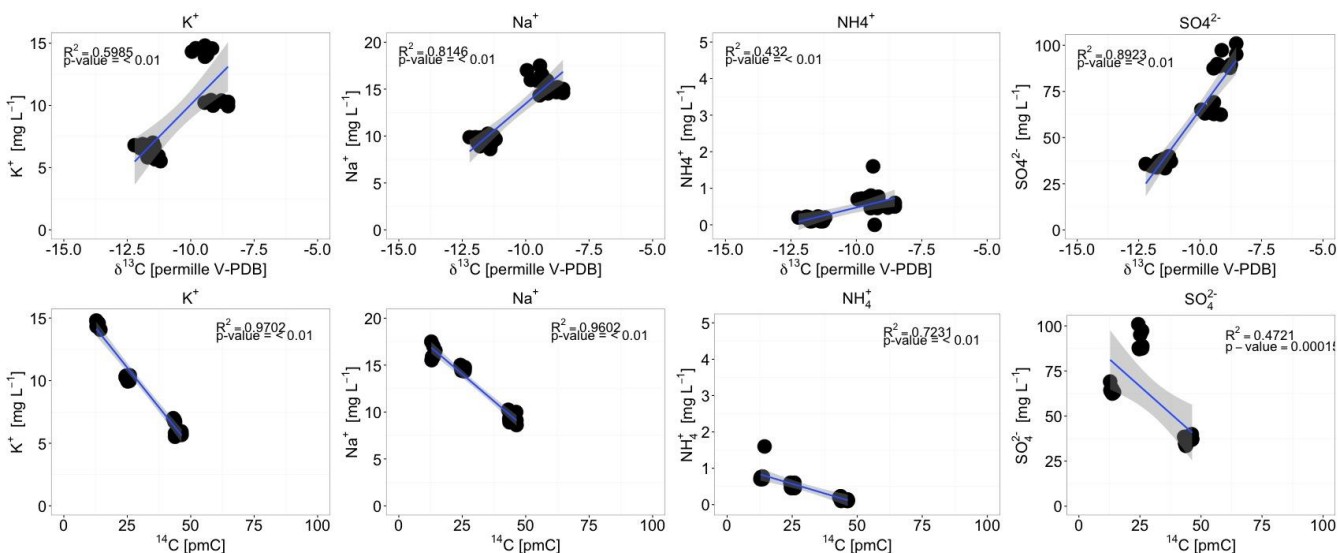

**Figure 7: Correlations between elemental concentration in group 2 and 3 wells and measured δ¹³C_DIC and ¹⁴C_DIC values.**