# Peer review of "Carbon isotopes of dissolved inorganic carbon reflect utilization of different carbon sources by microbial communities in two limestone aquifer assemblages"

_Hydrology and Earth System Sciences, 2016_

## Referee Comment (RC1) · Anonymous Referee #1 · 23 Dec 2016

*General remarks*

The manuscript describes the results of an isotope study ($\delta$13C, 14C) in the Hainich Critical Zone Exploratory that tried to evaluate flow paths in two aquifers systems by inverse chemical modeling.

The manuscript is adequately organized and contains no grammatical or orthography errors. Text is concise and to the point. The English is fine and does not need any revision.

The introduction should more clearly state the objectives of the study. It remains some-

what unclear what exactly was the aim(s) of the research.

With respect to the conclusions, it remains also unclear if the results obtained at the HCZE can be transferred to other layered karstic aquifer systems on a regional or global scale. The conclusion needs a bit of an outlook and more global perspective at the end. This holds also true for the importance of groundwater resources. At current state, the conclusions are only a summary of the discussion.

The tables were obviously not thoroughly prepared; units are completely missing and Table captions are very brief with most relevant information missing. This needs definitely improvement.

Overall, I would recommend this study for publication in HESS with some minor revisions outlined below.

*Specific comments*

P1L27 dramatically => considerably/significantly

P3L1 If traditional corrections have been applied – what are then untraditional models? To my knowledge the review by Han and Plummer lists nearly every correction that was published for 14C.

P3L28 To be more clear please add: "according to the stoichiometry of Eq. (1)

End of the introduction: The authors clearly state what they did but not so clearly why. I encourage the authors first to formulate one/some objectives of the study and afterwards mention briefly the applied methods.

P6L10 Any modifications to the cap to make it "gas tight" or the standard blue color Duran bottle caps?

P6L20 Please mention the pore size ($0.45\mu$m?) and procedure for filtering to distinguish between DOC and TOC.

[Figure]

Interactive
comment

P6L29 How was the sample extracted from the 1L Schott bottle since contact with atmospheric $CO_2$ might alter the $\delta13C$ of the sample.

P6L30 Sure they were flushed with N2 not helium? Double-check here.

P7L14 in fact none of the three materials used is rubber, in the sense of butyl rubber (silicone, PTFE and Viton/FKM)? So where does the contamination comes from?

P8L1 pmC => pMC ? (you might also use pmC but be consistent throughout the text); check also on eq. (5), P9L22 etc.

P8L21 here you might simply refer to eq. (4) ?

P8L22 Please be more precise here. Do you mean that the error is the precision of the analysis sequence (e.g. based on the repeated analysis of a control or drift sample)?

P12L1 note that this is the Cambridge half-life time. A (activity) and t (time) should be italic characters.

P13L2 How can you report a value with two digits if your precision is $\pm0.3‰$ (P7L7)? Also, if your s.d. is $\pm0.19‰$ it does make any sense to report a value of $-11.66\pm0.19‰$ Please change to $(-11.7\pm0.2)‰$ . Also check numbers in other lines here.

P13L7 For pMC the same applies as for the precision of $\delta13C$. How can you report a value of $62.23\pm7.01$ pMC? Your s.d. is by far larger than your reported value. I strongly suggest changing these values to $(62\pm7)$ and $(13.4\pm0.5)$ pMC.

Same applies for DOC (section 3.3) and 3.4

P15L28 change in preparation to unpublished (might be subject to change latter in the production process)

P16L12ff Did the authors measure 3H activities in their samples? This might provide some more information of the presence of young waters (i.e. younger than 60 years).

*Tables*
Table 1

Units are missing. You cannot report a value without its unit. Also use $\mu$mol (not umol; Fe and NH4+). On what is the s.d. ($\pm x$) values based? Please include a brief description to the Table caption. Also mention in Table captions full names of HTL and HTU.

Table 2

Units are missing (‰ vs V-PDB and pMC) from the first line. See my comments for Table 1 and also my comment on reporting a value with a rather large uncertainty (P13L2). It simply makes no sense to report a value of (65.16$\pm$4.03) => change to (65$\pm$4) or (65.2$\pm$4.0).

Table 4

See my comments in units above. Provide a citation for Tamers and F&G or at least mention in the Table captions that these are explained in the review by Han and Plummer.

*Figures*

Figs. 2, 3, 6 and 7:

Data points are hard to distinguish if the plot close or over each other. I suggest using a thin black line for the circles. In Fig 7 reduce symbol size and use open circles instead of filled black to improve readability.

Fig 6b and 6c

- What is the unit of DIC on x-axis?

- Figure caption of 6b is wrong: The caption states that 14C is plotted vs 1/DIC but y-axis is labeled $\delta$13C.

- Figure caption of 6c is wrong: The caption states that $\delta$13C is plotted vs 1/DIC but

y-axis is labeled pmC. Note that $\delta$13C is not a concentration.

*Technical comments*

P5L10 check parentheses (H1 (upslope

P6L2 table 1 => Table 1?

P7L18 space character ( 0 pMC)

P15L29 vales => values

P18L13 semicolon?

---

## Referee Comment (RC2) · Anonymous Referee #2 · 15 Feb 2017

General Comments This manuscript investigated the carbon isotopes of DIC, DOC and POC in two limestone aquifer assemblages in the Hainich Critical Zone Exploratory (CZE). The authors used a variety of techniques to study the evaluation of DIC carbon isotopes. The authors did a lot of works, conducted a very nice statistic analysis and made a beautiful figure. The paper reveals some interesting findings. The paper is also well written (in terms of structure and English both). In my opinion, the paper deserves publication with some minor corrections. Specific Comments 1) I would encourage authors to rewrite the objectives. Give a clear message to the reader what you did. Unnecessary description makes the text a bit boring. The authors are expected to

concise introduction. 2) The locations of the CZE is more important to understand this paper. I would suggest the authors to add a regional geological map. 3) It was also interesting to see that the influence of land use and climate change on groundwater compositions and DIC.

---

## Author Comment (AC1) · 15 Apr 2017

Dear referee,

Thank you for handling our manuscript. I would like to send you answers to your comments. The comments were constructive and helpful. We improved our manuscript where it was necessary and clarified and deispeled reservations where it was possible.

General comment 1: We tried to improve our introduction and outline more clearly the objectives of our study.

General comment 2: The conclusions are based on the results we obtained from the HCZE. We think that our results can be extrapolated to other aquifers, which have a similar geological and geochemical setting. This is clarified in the text.

General comment 3: Table and table captions were improved.

Specific comments:

P1L27: Ok

P3L1: Ok, the word traditional was removed

P3L28: Ok

End of introduction: We stated the objectives in the introduction, which were to use stable and radioactive carbon isotopes to evaluate processes that influence carbon cycling within our aquifer transect. We additionally used molecular techniques to identify microbial key players that contribute to carbon cycling. We hypothesised that $CO_2$ fixing microorganisms contribute to carbon cycling and this is reflected in DIC isotopes. This is in our opinion clearly stated in the text.

P6L10: We used standard blue colour Duran bottle caps with inlay rings which provide gas tightness.

P6L20: Ok

P6L29: The sample was extracted from the 1L schott bottle with a gasthight syringe in a glove bag filled with $N_2$ and added to the vials via a butyl rubber septum.

P6L30: Yes, they were flushed with $N_2$

P7L14: The source of contamination is not so relevant, because we used blanks to correct for it.

P8L1: Ok

P8L21: Ok

P8L22: Ok

P12L1: Ok

P13L2: Ok, digits were adjusted P13L7: Ok, this was adjusted

P15L28: Ok

P16L12ff: 3H was not measured, but it is planned in future.

Tables:

Table 1: Ok

Table 2: Ok

Table 4: Ok

Figures:

Figs. 2, 3, 6 and 7: The figures could not be adjusted further.

Fig 6b and 6c: Ok

Technical comments:

P5L10: Ok

P6L2: Ok

P7L18: Ok

P15L29: Ok

P18L13: Ok

---

## Author Comment (AC2) · 15 Apr 2017

Dear referee,

Thank you for handling our manuscript. I would like to send you answers to your comments. The comments were constructive and helpful. We improved our manuscript where it was necessary and clarified and dispelled reservations where it was possible.

Comment 1: The objectives were clarified in the introduction. We think that the descriptions of $CO_2$ chemistry in the aquifer are necessary to understand the overall scope of the paper for readers which are not familiar with the topic.

[Figure]

Comment 2: The geological setting of the area is precisely described in the paper of Kohlhepp et al., which is cited several times in our manuscript. The reader is referred to this paper in order to get a better insight to the geological setting of our study site. We think therefore that it is not necessary to include a geological map to our manuscript.

Comment 3: We discussed the potential impact of land use on DIC isotopes in our aquifer (P19L23ff). However, our conclusions remain somewhat hypothetical, because of a lack of data. More data, like 3H measurements would help, because they could give more information about the content of young waters in aquifers with low 14C values. However, such measurements could not be conducted in our study and remain a task for the future investigations.

―――――――――――――――――――――

---

## Author Response (AR1)

Answer to comments of reviewer #1

The introduction should more clearly state the objectives of the study. It remains someC1 HESSD Interactive comment Printer-friendly version Discussion paper what unclear what exactly was the aim(s) of the research

Ok, this was clarified in the text (P2L26)

P1L27 dramatically => considerably/significantly

Ok, this was changed

P3L1 If traditional corrections have been applied – what are then untraditional models? To my knowledge the review by Han and Plummer lists nearly every correction that was published for 14C.

Ok, this was changed

P3L28 To be more clear please add: "according to the stoichiometry of Eq. (1)

Ok, this was changed

End of the introduction: The authors clearly state what they did but not so clearly why. I encourage the authors first to formulate one/some objectives of the study and afterwards mention briefly the applied methods.

A paragraph, which states the main objectives of our study was added (P4L19)

P6L10 Any modifications to the cap to make it "gas tight" or the standard blue color Duran bottle caps?

We used standard blue colour Duran bottle caps with inlay rings which provide gas tightness.

P6L20 Please mention the pore size (0.45μm?) and procedure for filtering to distinguish between DOC and TOC.

Ok

P6L29 How was the sample extracted from the 1L Schott bottle since contact with atmospheric CO2 might alter the $\delta^{13}C$ of the sample.

The sample was extracted from the 1L schott bottle with a gasthight syringe in a glove bag filled with N2 and added to the vials via a butyl rubber septum.

P8L1 pmC => pMC ? (you might also use pmC but be consistent throughout the text); check also on eq. (5), P9L22 etc.

Ok

P8L21 here you might simply refer to eq. (4) ?
Ok

P8L22 Please be more precise here. Do you mean that the error is the precision of the analysis sequence (e.g. based on the repeated analysis of a control or drift sample)?

Ok

P12L1 note that this is the Cambridge half-life time. A (activity) and t (time) should be italic characters.

Ok

P13L2 How can you report a value with two digits if your precision is ±0.3‰ (P7L7)? Also, if your s.d. is ±0.19‰ it does make any sense to report a value of -11.66±0.19‰ Please change to (-11.7±0.2)‰ . Also check numbers in other lines here.

Ok

P13L7 For pMC the same applies as for the precision of δ13C. How can you report a value of 62.23±7.01 pmC? Your s.d. is by far larger than your reported value. I strongly suggest changing these values to (62±7) and (13.4±0.5) pMC. Same applies for DOC (section 3.3) and

Ok

P15L28 change in preparation to unpublished (might be subject to change latter in the production process)

OK

P16L12ff Did the authors measure 3H activities in their samples? This might provide some more information of the presence of young waters (i.e. younger than 60 years).

3H was not measured, but it is planned in future.

Table 1
Units are missing (‰ vs V-PDB and pMC) from the first line. See my comments for Table 1 and also my comment on reporting a value with a rather large uncertainty (P13L2). It simply makes no sense to report a value of (65.16±4.03) => change to (65±4) or (65.2±4.0).

Ok

Table 2 Units are missing (‰ vs V-PDB and pMC) from the first line. See my comments for Table 1 and also my comment on reporting a value with a rather large uncertainty (P13L2). It simply makes no sense to report a value of (65.16±4.03) => change to (65±4) or (65.2±4.0).

Ok

See my comments in units above. Provide a citation for Tamers and F&G or at least mention in the Table captions that these are explained in the review by Han and Plummer.

Ok

Figures* Figs. 2, 3, 6 and 7: Data points are hard to distinguish if the plot close or over each other. I suggest using a thin black line for the circles. In Fig 7 reduce symbol size and use open circles instead of filled black to improve readability.

Figures will be kept as they are

Fig 6b and 6c - What is the unit of DIC on x-axis? - Figure caption of 6b is wrong: The caption states that 14C is plotted vs 1/DIC but y-axis is labeled δ13C. - Figure caption of 6c is wrong: The caption states that δ13C is plotted vs 1/DIC but C4 HESSD Interactive comment Printer-friendly version Discussion paper y-axis is labeled pmC. Note that δ13C is not a concentration.

Ok

Answer to comments of reviewer #2

Comment 2: The geological setting of the area is precisely described in the paper of Kohlhepp et al., which is cited several times in our manuscript. The reader is referred to this paper in order to get a better insight to the geological setting of our study site. We think therefore that it is not necessary to include a geological map to our manuscript.

Comment 3: We discussed the potential impact of land use on DIC isotopes in our aquifer (P19L23ff). However, our conclusions remain somewhat hypothetical, because of a lack of data. More data, like 3H measurements would help, because they could give more information about the content of young waters in aquifers with low 14C values. However, such measurements could not be conducted in our study and remain a task for the future investigations.